# Diffusion-based dynamics as a cognitive model of human speech production

## Abstract

Human language production requires transforming abstract communicative intent into fluent speech, yet the algorithmic nature of this transformation remains less understood. Most studies aligning large language models (LLMs) with brain activity have focused on autoregressive LLMs (aLLMs), which generate text left-to-right by committing to the next token. While effective at predicting neural and behavioral signatures of comprehension, this paradigm assumes incremental generation. In contrast, diffusion LLMs (dLLMs) construct sentences by iteratively denoising global representations. Despite their distinct generative dynamics, dLLMs now rival aLLMs on standard NLP benchmarks, prompting the question of whether the brain likewise engages in global, iterative refinement—especially during pre-articulatory planning when sentence structure remains flexible. To test this hypothesis, we correlated intermediate denoising steps of a dLLM with electrocorticography (ECoG) activity during naturalistic speech production. dLLM representations explained significant neural variance from pre- to post-production, with especially strong encoding in middle/inferior temporal and motor-related regions. These results support iterative refinement as a plausible neural mechanism of human speech planning.

## 1 Introduction

Human language processing—from comprehension to internal formulation to overt production—is a window into the mind's generative machinery. Most large language models (LLMs) used in model-brain alignment studies operate via a left-to-right, next-token prediction paradigm (Caucheteux et al., 2023; Gao et al., 2025; Goldstein et al., 2022; 2025; Schrimpf et al., 2021; Toneva & Wehbe, 2019; Antonello et al., 2024; Jain & Huth, 2018; Tang et al., 2023; **?**). These autoregressive architectures have proven surprisingly effective at capturing aspects of human brain activity during naturalistic language tasks, especially when scaled up (Gao et al., 2025; Antonello et al., 2024; Hong et al., 2024). However, they instantiate one specific algorithmic hypothesis about how linguistic output is constructed: sequential conditional commitment to the next word. Emerging diffusion LLMs (dLLMs), such as LLaDA (Nie et al., 2025) and Dream (Ye et al., 2025a), propose a qualitatively different generative mechanism. Instead of predicting the next token, they begin from a noisy, underspecified representation and iteratively denoise toward a coherent sentence. Despite their contrasting dynamics, large-scale dLLMs now rival aLLMs such as LLaMA3 (Grattafiori et al., 2024) and Qwen2.5 (Yang et al., 2024) across a range of NLP benchmarks. This computational plurality challenges the assumption that next-word prediction is the sole viable substrate for language modeling.

The question arises: Could the brain's language production resemble an iterative refinement more than a left-to-right sequence? In this view, speakers hold a graded, probabilistic proto-utterance that is progressively refined before speech onset, with lexical, syntactic, and discourse constraints gradually resolving into a coherent plan for articulation. Here, we test this hypothesis by correlating intermediate embeddings of dLLMs (LLaDA and Dream) across denoising steps with electrocorticography (ECoG) activity during naturalistic speech comprehension and production. We then examined whether human language production may more closely resemble the predictions of dLLMs than those of aLLMs. If so, this would suggest that human language production may not rely solely on autoregressive mechanisms as stated in the prior model-brain alignment literature (Antonello et al., 2024; Caucheteux & King, 2022; Schrimpf et al., 2021; Gao et al., 2025; Goldstein et al.,

2022; 2025; Toneva & Wehbe, 2019; Hong et al., 2024; Jain & Huth, 2018; Caucheteux et al., 2023; Tikochinski et al., 2025; Goldstein et al., 2024; Tang et al., 2023; d'Ascoli et al., 2024; Ye et al., 2025b). To the best of our knowledge, no published study has yet reported direct comparisons of a dLLM's internal representations with neural recordings during language tasks. We find that (1) earlier diffusion steps preferentially generate high-frequency content words, diverging from the sequential patterns in autoregressive models; (2) dLLM embeddings evolve in parallel with neural dynamics in temporal and motor regions; and (3) earlier steps align with pre-articulatory activity, while later steps align with motor execution and post-articulatory processing. Together, these results suggest that diffusion models may capture some aspects of the dynamics of human speech production in addition to autoregressive models.

## 2 RELATED WORK

**Diffusion LLM.** Early diffusion models such as Diffusion-LM (Li et al., 2022b) and D3PM (Austin et al., 2021) were relatively small, but they established the foundation for today's billion-parameter models such as LLaDA-8B (Nie et al., 2025) and Dream-7B (Ye et al., 2025a). LLaDA adopts the core architecture of LLaMA3. It uses the same tokenizer and Transformer layer stack as LLaMA3-8B, enabling direct performance comparisons. Crucially, LLaDA removes the causal (left-to-right) attention mask used in LLaMA models, allowing bidirectional self-attention over the sequence. A special masking token (<MASK>) is introduced into the vocabulary to represent "noisy" or corrupted tokens during diffusion-style generation. Dream's architecture is directly derived from Qwen2.5-7B and was initialized with Qwen2.5-7B's pretrained weights to bootstrap its knowledge. The key architectural modification for Dream was analogous to LLaDA's: switching from Qwen's causal masking to full bidirectional self-attention.

Both LLaDA and Dream depart from conventional autoregressive training in that they do not use next-word prediction on a prefix. Instead of producing one token at a time, they predict many tokens at once given a partially masked context. This means that the loss is computed over multiple token positions simultaneously (all masked tokens) rather than only the next position. The training data for dLLMs also must include very high masking ratios (up to 100% masked) so that the model learns to generate whole sequences from nothing. Another important difference is the use of time-step conditioning in dLLMs: the model is aware of a "step" or mask level during training, which aLLMs do not require. This was implemented by adding an encoding of the mask fraction or diffusion step index into the model's input or hidden layers. The outcome is that dLLMs learn a sequence of denoising steps rather than a single-step distribution. Apart from LLaDA and Dream, industry efforts, such as Mercury (Labs et al., 2025) and Gemini Diffusion (Deepmind, 2024) report generation speeds of thousands of tokens per second using optimized parallel sampling, demonstrating that dLLMs are becoming practical alternatives rather than mere academic curiosities.

**Model-brain alignment during language use.** In recent years, numerous studies have reported parallels between LLMs and human brain activity during language processing (Antonello et al., 2024; Caucheteux et al., 2023; Gao et al., 2025; Hong et al., 2024; Jain & Huth, 2018; Goldstein et al., 2022; 2025; Schrimpf et al., 2021; Toneva & Wehbe, 2019). For instance, GPT-2's word probabilities explained unique variance in ECoG responses in language areas, suggesting that both the brain and LLMs rely on predictive representations (Goldstein et al., 2022). More recent LLMs such as LLaMA (Touvron et al., 2023) and OPT (Zhang et al., 2022) have been shown to align more closely with brain activity during language processing, exhibiting a scaling law whereby larger models yield improved brain predictivity (Antonello et al., 2024; Hong et al., 2024; Gao et al., 2025). Although some non-autoregressive models (e.g., BERT, Whisper) have also been used to investigate human language comprehension (Toneva & Wehbe, 2019; Schrimpf et al., 2021; Caucheteux et al., 2021; Lamarre et al., 2022; Schwartz et al., 2019) and production (Goldstein et al., 2025), today's state-of-the-art LLMs are largely autoregressive. Correspondingly, the mainstream view in psycholinguistics holds that readers and listeners actively anticipate upcoming words in an autoregressive manner (De-Long et al., 2005; Kutas & Hillyard, 1980).

Yet the human brain goes beyond local, word-by-word prediction. Caucheteux et al. (2023) demonstrated that models trained to anticipate not only the next word but also upcoming words or sentence-level features better matched neural activity. dLLMs provide a natural test of this idea: by iteratively refining an entire sequence, they inherently generate predictions with a broader temporal and struc-

tural horizon than next-word generators. This iterative, global approach might be a better parallel for how we plan utterances, a process hard to emulate with purely left-to-right generation.

## 3 METHODS

### 3.1 EXTRACTING ECoG DATA DURING SPEECH COMPREHENSION AND PRODUCTION

Our ECoG data were drawn from a previously published study (Goldstein et al., 2025) comprising continuous 24/7 recordings from four patients (see Table 1 in Appendix A) who engaged in spontaneous conversations with family, friends, doctors, and hospital staff during their week-long stay in the epilepsy monitoring unit. Across patients, neural signals were collected from 675 intracranial electrodes. We selected 466 electrodes located in six left-hemisphere regions of interest (ROIs) defined by the "HCPMMP1_combined" atlas (Glasser et al., 2016): superior temporal gyrus (STG: 100 electrodes), middle and inferior temporal lobe (MTL/ITL: 89 electrodes), inferior frontal gyrus (IFG: 84 electrodes), dorsolateral prefrontal cortex (DLPFC: 55 electrodes), motor cortex (MC: 97 electrodes) and angular gyrus / temporo-parietal-occipital junction (AG/TPOJ: 41 electrodes). These regions have been shown to play critical roles in language use (Malik-Moraleda et al., 2022). All conversations were transcribed, and each word was time-aligned with the concurrent ECoG signals. After preprocessing (see Appendix B for details), we divided the dataset into comprehension (listening) and production (speaking) periods, yielding 50 hours (289,971 words) of comprehension data and 50 hours (230,238 words) of production data in naturalistic settings. We also divided sentences into shorter (5-25 words) and longer groups (25-50 words) to examine whether sentence length influences model encoding performance across aLLMs and dLLMs (see Figure 17 in Appendix O for distribution of sentence length in words and duration.) We sampled ECoG activity at five evenly spaced points within each utterance, together with five points from the two seconds preceding and following production, yielding a 15-timepoint ECoG time course for each utterance (see Figure 1). Unlike prior studies (Goldstein et al., 2022; 2024; 2025) that focus on word-level alignment, our analysis centers on entire sentences. Sampling five evenly spaced steps allows us to standardize the number of timepoints per sentence while directly addressing our research question: whether earlier denoising steps of diffusion models align with earlier neural time windows, and whether later steps align more strongly with later time windows.

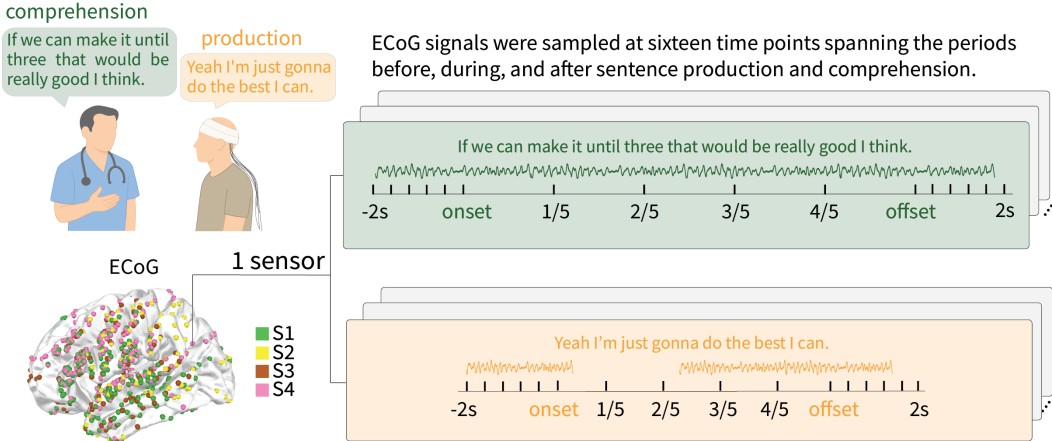

Figure 1: Extracting ECoG data during speech comprehension and production. ECoG activity at five evenly spaced points within each utterance were sampled, together with five evenly-spaced points from the two seconds preceding and following production, yielding a 15-timepoint ECoG time course for each utterance.

### 3.2 EXTRACTING SENTENCE EMBEDDINGS ACROSS STEPS

We selected LLaDA-8B (Nie et al., 2025) and Dream-7B (Ye et al., 2025a) to test their alignment with brain activity during speech comprehension and production. We also included their autore-

gressive counterparts LLaMA3-8B (Grattafiori et al., 2024) and Qwen2.5-7B (Yang et al., 2024) for comparison (see Table 2 in Appendix C for model details). We first extracted sentence embeddings from every layer of the four LLMs (averaged over all tokens) and performed encoding analyses for each layer at every sensor within the selected ROIs and at every timepoint for each subject. We then averaged the resulting correlation coefficients across sensors and timepoints for each layer. We found that for comprehension sentences, the best-performing layers for LLaMA, LLaDA, Qwen, and Dream were Layers 15, 12, 13, and 18, respectively (see the dotted lines in Figure 5 in Appendix D). For production sentences, the corresponding best layers were 30, 27, 14, and 27 (see the solid lines in Figure 5 in Appendix D). Across all models, the best layers for comprehension tend to occur earlier than those for production. Based on these findings, we extracted embeddings across the five progressive generation steps from the best-performing layer of each LLM.

For aLLMs (LLaMA, Qwen), we divided the sentence into five progressively longer partial inputs, containing roughly the first 20%, 40%, 60%, 80%, and 100% of words. For example, for the sentence "So maybe you did have something which is infinitely better than zero," the five partial inputs would be: "So maybe", "So maybe you did", "So maybe you did have something", "So maybe you did have something which is infinitely", "So maybe you did have something which is infinitely better than zero". Within each step, we averaged the token embeddings across all words included in that segment to obtain a single stage-level representation. The resulting layer-wise representations consist of five progressive embeddings per sentence per model layer, capturing how meaning representations evolve as linguistic context accumulates (see Figure 2, upper panel). For dLLMs (Dream, LLaDA), we first obtained the diffusion order of the original sentence (e.g., "have is So which something maybe did you zero infinitely better than"), and then extracted the embeddings for each of the five denoising steps by averaging the token embeddings across the entire sentence at that step (see Figure 2, lower panel). Importantly, the hidden states of unrevealed tokens continue to update at every step and gradually converge toward their final representations, allowing the denoising trajectory to capture the progressive refinement characteristic of diffusion-based generation. In this way, the stepwise embeddings for both aLLMs and dLLMs reflect how meaning is represented as it unfolds—either through left-to-right generation or through iterative denoising. Below we outline the algorithm used to extract sentence embeddings across generative steps from dLLMs:

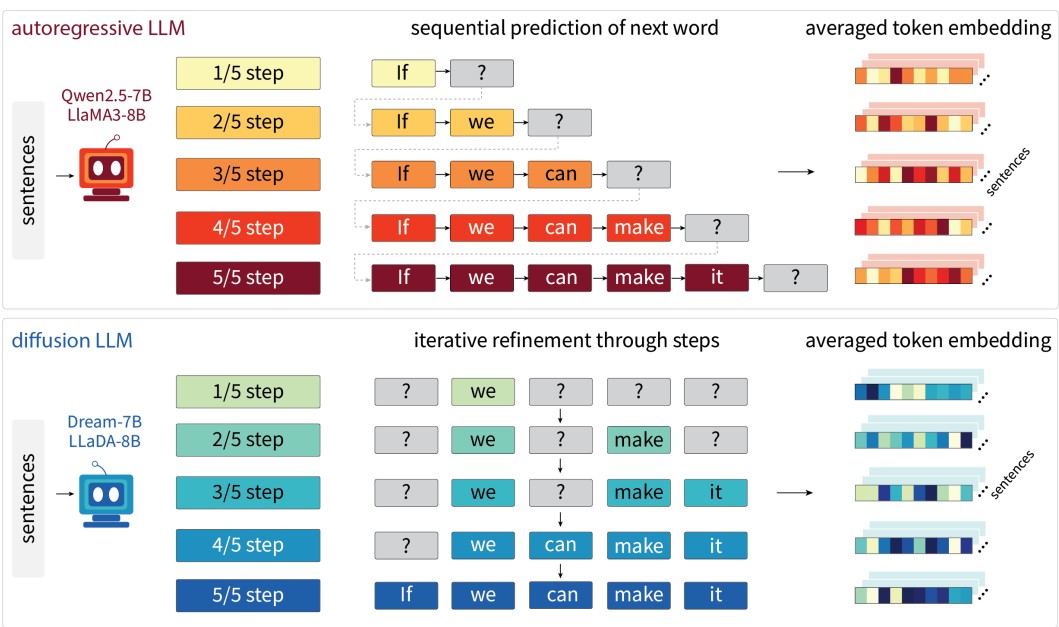

Figure 2: Extracting sentence embeddings at 5 generation steps in diffusion and autoregressive LLMs.

**Input preparation**. Each target sentence was embedded within a conversational prompt (e.g., "In a casual conversation, you heard 'comprehension' and you responded 'production'"). The compre-

hension field was populated with the preceding utterance from another speaker (see Appendix E for examples). Model forward passes were then executed, and hidden representations from the best-performing layer were extracted for the target tokens. To initialize the diffusion process, all target tokens were replaced with a designated mask symbol ([gMASK] for LLaDA and ¡|mask|¿) yielding a fully masked response sequence appended to the prompt.

**Initial position selection.** The fully masked sequence was passed through the model. For every masked position, we computed the softmax probability of the correct target token. The position with the highest confidence score was selected as the first revealed token. This ensured that the diffusion process started from the location where the model was most certain of the ground-truth content given only contextual cues.

**Iterative revelation.** Following this initialization, the model iteratively revealed one additional token per step. At each iteration, previously revealed tokens were fixed in place, while unrevealed positions remained masked. A forward pass generated logits for all masked positions. For each position, the probability assigned to its correct token was extracted. The position with the highest confidence was then revealed and added to the growing set of fixed tokens. Thus, each diffusion model sentence embedding at Step $k$ represents a state where the model has confidently placed $k$ out of $n$ words in their positions, while the remaining words are still masked. This greedy loop continued until all tokens in the target sentence had been revealed.

**Order recording.** The full revelation sequence was stored as an ordered list of positions, where each index indicated the step at which a token was revealed. In practice, this list was aligned to the original tokenization, such that each word could be assigned to the earliest step among its constituent tokens. This produced a word-level revelation trajectory reflecting the model's progressive reconstruction dynamics. Based on the diffusion sequence, each sentence was divided into five steps—20%, 40%, 60%, 80%, and 100% of words (see Figure 2, lower panel).

**Greedy confidence principle.** The algorithm implements a greedy search strategy: at each step, the most confident masked position is revealed, with no backtracking. Although not globally optimal, this procedure is computationally efficient and reflects a psychologically plausible mechanism of iterative refinement under uncertainty. The algorithm for token revelation in dLLMs is summarized below:

---

**Algorithm 1** Greedy confidence-based token revelation in dLLMs

---

**Require:** Model $M$, Tokenizer $T$, Context $c$, Target $s$, Mask token id $m$
**Ensure:** Original tokens, Revelation order, Step indices
1: Construct prompt
2: $target\_ids \leftarrow T.encode(s)$
3: $L \leftarrow \text{length}(target\_ids)$
4: $current\_resp \leftarrow [m]^L$
5: Run $M$ on prompt + $current\_resp$
6: Compute $conf[p] = \text{softmax}(logits[p])[target\_ids[p]]$ for all $p$
7: $best\_pos \leftarrow \arg\max_p conf[p]$
8: Reveal $best\_pos$; update state and record order
9: **while** $|revealed| < L$ **do**
10:     Run $M$ on prompt + $current\_resp$
11:     **for** each $p$ not in revealed **do**
12:         $conf[p] \leftarrow \text{softmax}(logits[p])[target\_ids[p]]$
13:     **end for**
14:     $best\_pos \leftarrow \arg\max conf[p]$
15:     Reveal $best\_pos$; update state and record order
16: **end while**
17: **return** original tokens, revelation order, step indices

---

**Sentence embedding extraction.** After determining the revelation order of words in dLLMs, we computed sentence embeddings for each of the five steps by averaging token-level states at that step, yielding a single vector per step (dimension 3,584 for Dream and Qwen, 4,096 for LLaDA and LLaMA). The resulting arrays were stored in (n sentences) × (5 steps) × (embedding dimension). We used averaged aLLM embeddings across the five generation steps, rather than last-token em-

beddings, to ensure a fair comparison with the dLLM embeddings, which are themselves averaged across all tokens at each denoising step. In dLLMs, unrevealed tokens contribute "noisy" representations that gradually sharpen across steps; averaging therefore reflects the evolving whole-sentence representation. Using last-token embeddings for aLLMs but averaged embeddings for dLLMs would introduce an asymmetry in the comparison. We computed the correlation between last-token and averaged-token embeddings for each sentence at each step in the aLLMs. Although the correlation decreases as more tokens are included in the average at later steps, all correlations remain in the range of 0.6–0.8 (see Figure 6 in Appendix F), suggesting that the averaged embeddings still reasonably approximate the last-token embeddings. We further added PCA of the 5-step last-token embeddings from aLLMs and the encoding results using last token embeddings and the results did not differ much from averaged token (see Figure 15 and Figure ?? in Appendix N.

### 3.3 ALIGNING SENTENCE EMBEDDINGS WITH ECoG ACTIVITY

We modeled neural responses from multiple embedding sources using a banded ridge (multiple-kernel) regression (Dupré la Tour et al., 2022), implemented with Himalaya's `MultipleKernelRidgeCV`. Each embedding model (Dream, LLaDA, Qwen, LLaMA) was treated as a separate kernel with its own regularization parameter, enabling joint integration of embedding spaces while adaptively weighting their contributions. We also included two control variables: (1) the mean log-mel spectrograms of the audio for each sentence; (2) the sentence embeddings of the last layer of Whisper encoder (averaged over all tokens in a sentence). All features were reduced to the first 100 principal components. Details of banded ridge regression analyses were described in Appendix G.

### 3.4 CONTROL ANALYSES

We performed two additional control analyses: (1) We permuted the ECoG data to examine whether any spurious correlations existed between the LLM embeddings and the neural responses. The regressors entered into the banded ridge regression were the same: the spectrogram, the last encoder layer of the Whisper model, and the best-performing layers of LLaDA, LLaMA, Dream, and Qwen at each denoising or autoregressive step for each sentence. All features were reduced to the first 100 principal components. The dependent ECoG data were then permuted across sentences for all 15 timepoints, ensuring that the embeddings no longer corresponded to the correct sentence-level neural activity. (2) We randomized the revelation order of the dLLMs, such that the tokens revealed at each denoising step were no longer the most probable ones derived from the greedy confidence-based token revelation algorithm. This serves as a sanity check to determine whether an arbitrary revelation order would also produce significant correlations with the neural data.

### 3.5 ANALYZING WORD-LEVEL FEATURES ACROSS DIFFERENT GENERATION STEPS

**Visualizing embeddings with PCA.** We applied principal component analysis (PCA) to embeddings from dLLMs and aLLMs to visualize how sentence representations evolve across five steps. For each subject, embeddings of shape (n sentences $\times$ 5 steps $\times$ dimension) were concatenated and reshaped so each sentence–step pair was treated as a data point. PCA reduced dimensionality to three components, performed separately for diffusion and autoregressive LLMs, with explained variance averaged across models.

**Measuring diffusion–autoregressive distances.** Representational differences were quantified using Jensen–Shannon (JS) divergence between 5-step embeddings from LLaMA vs. LLaDA and Qwen vs. Dream. At each step, sentence-level JS distances were computed for paired autoregressive-diffusion LLMs and averaged across the two pairs.

**Word frequency across steps.** Log word frequencies for words at each step were retrieved from the Google Books N-gram corpus (Google, 2010) for both model families and compared across steps using paired $t$-tests with FDR correction for multiple comparisons.

**POS distributions across steps.** Part-of-speech (POS) tags obtained with spaCy were grouped into four categories: NOUN (including PROPN, PRON), VERB (including AUX), ADJ/ADV, and

FUNC (all remaining tags). The percentage measures were computed as follows: for each generation step, we first calculated the proportion of each POS tag relative to the total number of tags at that step across all sentences, and then averaged these proportions across the two dLLMs and across the aLLMs, respectively. For example, if in Step 1 the dLLM ordering yields 50 nouns and the aLLM (i.e., sequential) ordering yields 40 nouns, and there are 100 total words in Step 1 across all sentences, then the percentage of nouns would be 50% for dLLMs and 40% for aLLMs. Statistical significance between dLLM and aLLM percentages were examined using paired $t$-tests with FDR correction for multiple comparisons.

## 4 RESULTS

### 4.1 WORD-LEVEL FEATURES ACROSS DIFFUSION AND AUTOREGRESSIVE STEPS

Table 3 in Appendix H presents five illustrative examples of words generated across the five diffusion steps. At Step 1 the most predictable token (often a content word like "dinner" or "her husband") is placed, by Step 5 the full sentence is formed. We also quantified the positional differences between the positions of words from dLLMs and their original positions in the sentence. The results showed that words from the first step are distributed relatively evenly across Steps 2–4 (see Figure 7 in Appendix H). Figure 3a shows the first three PCs of the 5-step embeddings for aLLMs and dLLMs. dLLMs exhibited a clearer temporal trajectory than aLLMs: embeddings from successive steps were more distinctly separated in principal component space. Moreover, the first PC of diffusion embeddings explained substantially more variance (25.2%) than that of autoregressive embeddings (9.7%), highlighting stronger step-wise differentiation in dLLMs. We further quantified these differences by computing Jensen–Shannon divergence between model families (Figure 3b), which revealed consistently greater separation at earlier steps.

Significant differences also emerged in the distribution of log word frequency across generative steps (Figure 3c). At the onset and offset of sentences, dLLMs produced words of significantly higher frequency than aLLMs (Step 1: $t = 16.21$, $p < 10^6$; Step 5: $t = 17.16$, $p < 10^6$). In contrast, in the middle portions of sentences, aLLMs favored higher frequency words (Step 3: $t = -10.74$, $p < 10^6$; Step 4: $t = -13.66$, $p < 10^6$). No significant difference was observed at Step 2 ($t = 1.86$, $p = 0.063$). These results reveal a U-shaped frequency pattern for dLLMs—relying on common, high-frequency words at sentence onset and offset while using relatively lower frequency words mid-sentence—whereas aLLMs show the opposite tendency in the middle steps.

Finally, POS distributions diverged systematically between model types (Figure 3d): Paired $t$-tests reveal consistent boundary effects: dLLMs favored more content words (NOUN, VERB) at Step 1 ($t = 6.08$, $p < 10^6$) and Step 4 ($t = 2.40$, $p = 0.017$), whereas aLLMs used more NOUNs at Step 5 ($t = -18.92$, $p < 10^6$). VERB and FUNC categories likewise exhibited mirrored trends: dLLMs used more verbs at sentence start (Step 1: $t = 13.18$, $p < 10^6$), while aLLMs dominated the later steps. ADJ/ADV showed smaller but significant differences in a few steps. The shifting POS proportions support the hypothesis that dLLMs reorganize lexical categories across the course of generation differently than aLLMs.

### 4.2 BRAIN ENCODING PERFORMANCE FOR SHORTER SENTENCES

#### 4.2.1 SPEECH PRODUCTION

Figure 4a illustrates the encoding performance of aLLMs during naturalistic speech production. Time-resolved encoding reveals consistently higher correlations across ROIs for later-step embeddings, suggesting that autoregressive embeddings become more predictive of neural activity as sentence production unfolds. dLLM embeddings exhibit a systematic temporal alignment with neural dynamics during speech production. Earlier diffusion steps correlated more strongly with neural activity in temporal regions such as STG and MTL+ITL, DLPFC, IFG and MC prior to articulation, suggesting that these representations capture pre-articulatory planning processes. By contrast, later diffusion steps achieved higher correlations in IFG and MC around and after sentence offset, consistent with mid- and post-articulatory stages of production (see Figure 4b). These findings suggest

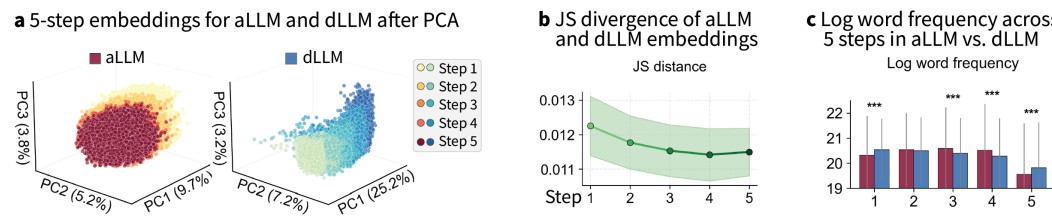

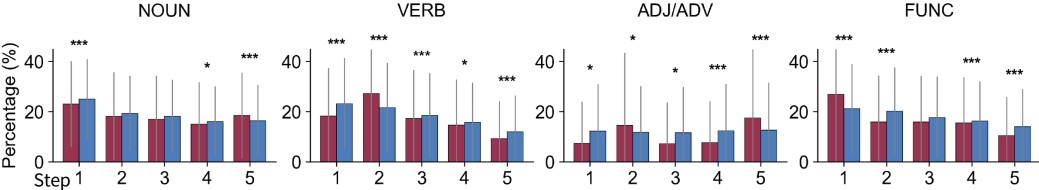

Figure 3: Structural and lexical differences across generative steps in aLLMs and dLLMs. **a** PCA shows greater step-wise separation in diffusion embeddings than in autoregressive ones. **b** Jensen–Shannon divergence decreases across steps. **c** dLLMs favor higher-frequency words at sentence boundaries. **d** dLLMs produce more content words early, whereas aLLMs favor function words.

that dLLMs not only capture overall neural dynamics but also differentiate between pre-articulatory planning and later motor-related processes across cortical regions.

### 4.2.2 SPEECH COMPREHENSION

For comprehension, we observed overall higher alignment between late-step aLLM embeddings and ECoG activity across ROIs and timepoints (see Figure 4c). This pattern is particularly pronounced in the DLPFC and AG/TPOJ. Late-step dLLM embeddings also best predict ECoG activity in a later time window near sentence offsets in the STG (see Figure 4d), similar to the pattern observed for aLLMs in STG.

Note that the encoding performance appear lower than those reported in prior studies (Goldstein et al., 2025). This is because we used banded ridge regression with all four LLMs included simultaneously, and we reported only the unique variance (r values) independently explained by each model. Given that the two aLLM embeddings and the two dLLM embeddings are highly correlated, the unique variance attributed to each individual model is necessarily reduced. Additionally, our encoding analyses were performed at the sentence level rather than the word level as in Goldstein et al. (2025), resulting in substantially fewer trials for training the encoding models. This reduced trial count may have contributed to the lower brain scores observed in our study.

We have also performed variance partitioning on dLLMs and aLLMs across the six ROIs. Specifically, for each step, we first conducted 2 ridge regressions, first with the two aLLM (LLaMA and Qwen) embeddings, then with the two dLLM embeddings (LLaDA and Dream). We then conducted a banded ridge regression with the 4 sets of embeddings. The shared variance is the difference between the correlation coefficients of the two ridge regressions and the banded ridge regression. We computed percentages of the shared variance and unique variance over the overall correlation coefficients of the banded ridge regression. The results are shown in Tables 4-8 in Appendix I.

### 4.2.3 ACOUSTIC AND SPEECH FEATURES.

For production, we observed higher alignment of Whisper embeddings in the middle-sentence position in the MTL/ITL and MC across all steps. For comprehension, Whisper also showed highest

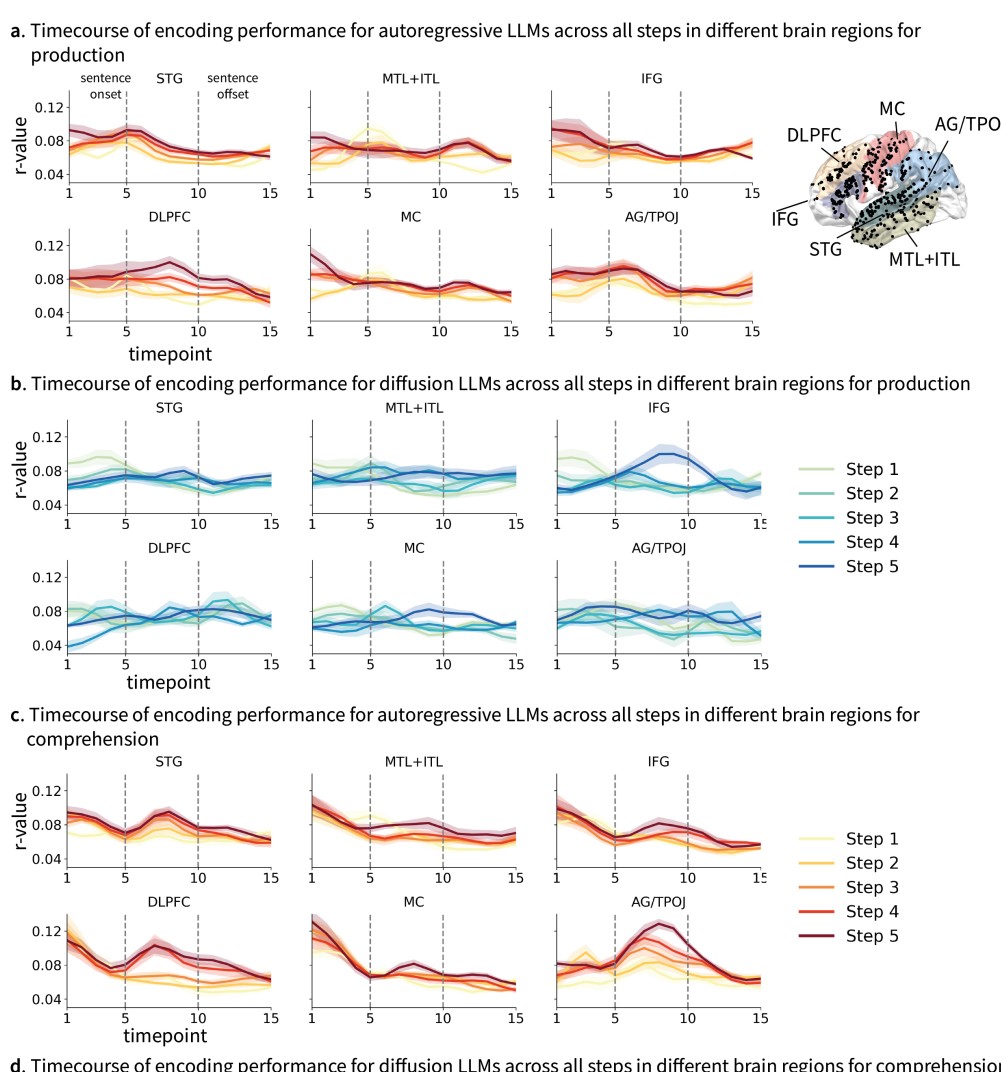

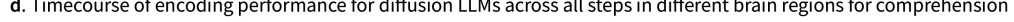

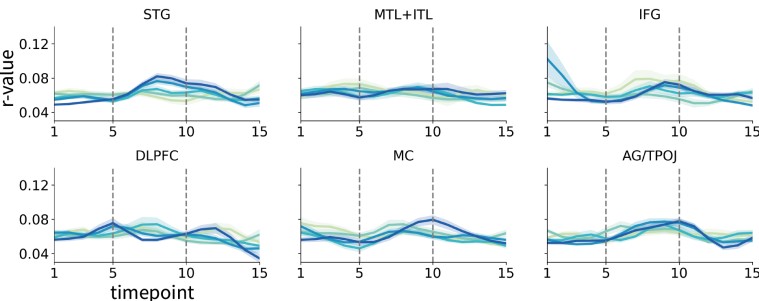

Figure 4: Encoding performance of aLLMs and dLLMs during speech production **a,b** and comprehension **c,d**. Sentence length is between 5-25 words. In this and all following line plots, shaded areas represent one standard error.

correlation in the middle-sentence position in the STG. Spectrogram does not show better model fit compare to other regressors in both production and comprehension (see Figures 8 and 9 in Appendix J).

### 4.2.4 CONTROL CONDITIONS.

We observed no significant correlations in any of the ROIs for either control condition—permuted ECoG data (see Figures 10 and 11 in Appendix K) or randomized revelation order of the dLLMs (see Figure 12 in Appendix K), suggesting that the observed model–brain alignments in the main analyses are not driven by random associations or arbitrary token revelation sequences.

We have also performed the same analyses on longer sentences (25-50 words) to examine whether sentence length could influence the encoding performance of aLLMs and dLLMs. We still found higher alignment of earlier step dLLM embeddings in the MTL and DLPFC at earlier time windows during production, while no such pattern has been observed in aLLM results (see Figure 13a,b in Appendix L. For comprehension, we observed no clear pattern of neural dynamics for both aLLMs and dLLMs (see Figure 13c,d in Appendix L.)

## 4.3 EXTENDING TO FMRI DATA DURING NATURALISTIC LISTENING

We extended our encoding analysis to an openly available fMRI dataset in which 49 English speakers listened to approximately around 100 minutes of naturalistic narrative (Li et al., 2022a). Due to the lower temporal resolution of fMRI, we performed the regression only at the sentence onset and offset (adding 5 seconds to capture the peak of the hemodynamic response). In this dataset, we observed overall higher alignment for last-step aLLM embeddings in IFG and for mid-step dLLM embeddings in MTL/ITL, but no consistent or interpretable temporal pattern emerged. These results further suggest that the diffusion-based representations are most relevant for production, whereas comprehension does not exhibit the same diffusion–brain alignment (see Figure 14 in Appendix M).

## 5 DISCUSSION AND CONCLUSION

This work provides the first direct evidence that dLLMs capture some aspects of the neural dynamics of human speech production in ways that qualitatively differ from aLLMs. Whereas aLLMs gradually increase neural predictivity as tokens accumulate, dLLMs show sharper step-wise differentiation: earlier diffusion steps align with pre-articulatory activity while later steps align with activity during and after articulation within middle/inferior temporal and motor cortices.

Note that it would be valuable to evaluate encoding performance on sentences containing unlikely continuations that aLLMs typically fail to capture. However, our current dataset consists exclusively of naturalistic conversations and therefore does not include experimentally manipulated or systematically constructed unlikely outcomes. That said, spontaneous conversational speech in our dataset does contain numerous hesitations, repairs, and speech errors (e.g., "really it feels a lot better, they took the, they took the thing out, the drain out."), which are inherently low-probability and generally not produced by LLMs during free generation. Thus, we believe the naturalistic data already offer many instances of disfluent or unlikely continuations that LLMs struggle to predict. Future work that systematically probes generative performance under controlled improbable scenarios would further strengthen the conclusions.

Several caveats remain: First, our token revelation procedure reflects one particular instantiation of diffusion dynamics; alternative denoising strategies may produce different mappings. Secondly, speech production relies on both acoustic and auditory paths alongside language production, yet we only focused on text-based models due to the practical limitation that there are no openly-available diffusion-based speech LLMs. Lastly, higher model–brain alignment does not necessarily imply that a model definitely implements the same mechanisms as the human brain; Rather, the stronger fit observed for diffusion LLMs during production suggests only that the model captures certain statistical or representational properties that are more predictive of the neural responses within the specific task and dataset examined.

In conclusion, our study shows that dLLMs could potentially explain some aspects of human speech production. This work opens several avenues: testing dLLM–brain alignment with other modalities (fMRI, MEG) during language production, exploring finer-grained layer-wise dynamics, and developing hybrid models that integrate sequential and diffusion principles.

## ETHICS STATEMENT

Participants gave informed consent in accordance with protocols approved by the XXX University Institutional Review Board. They were explicitly informed that participation was independent of their clinical care and that withdrawal would not affect their medical treatment. We do not anticipate any additional ethical concerns.

## REPRODUCIBILITY STATEMENT

We provide detailed descriptions of the procedures for extracting sentence embeddings and implementing banded ridge regression encoding models. All computations were performed using standard Python packages, including Hugging Face for language model handling and MNE-Python for neural data processing. The code and associated data for replicating the encoding analyses will be released publicly upon acceptance of the paper.

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

## A   PARTICIPANTS

Table 1: Patient demographics and clinical characteristics.

|  | **P1** | **P2** | **P3** | **P4** |
|---|---|---|---|---|
| Age (years) | 53 | 26 | 48 | 24 |
| Sex | F | M | F | M |
| Electrodes implanted | 104 | 125 | 255 | 192 |
| Hours of speech recorded | 17 | 37 | 17 | 29 |
| Words recorded | 79,654 | 213,473 | 117,800 | 109,282 |
| Comprehension words | 47,642 | 109,967 | 71,754 | 60,608 |
| Production words | 32,012 | 103,506 | 46,046 | 48,674 |
| Pathology / Seizure focus | Posterior temporal lobe (neocortical) epilepsy; seizure focus adjacent to posterior temporal lesion | Left anteromedial temporal lobe epilepsy | Right anteromedial temporal lobe epilepsy; ictal onsets localized to temporal pole and hippocampus | Focal epilepsy in left hemisphere with broad focus including temporal neocortex, frontal operculum, postcentral gyrus, insula |
| Implant | Left grid, strips, depth | Left grid, strips, depth | Bilateral strips/depths, left grid | Left grid, strips, depth |

## B   ECoG PREPROCESSING

The ECoG preprocessing pipeline mitigated artifacts arising from movement, faulty electrodes, line noise, abnormal physiological signals (e.g., epileptic discharges), eye blinks, and cardiac activity. A semi-automated procedure was used to identify and remove corrupted data segments (e.g., seizures, loose wires), while additional noise was attenuated using fast Fourier transform (FFT), independent component analysis (ICA), and de-spiking methods. Neural signals were then band-pass filtered in the broadband range (75–200 Hz), and the power envelope was computed as a proxy for each electrode's average local firing rate. The resulting signals were z-scored, smoothed with a 50-ms Hamming kernel, and trimmed by 3,000 samples at each end to minimize edge effects. All preprocessing was conducted using custom MATLAB 2019a (MathWorks) scripts.

## C   MODEL DETAILS

Table 2: Model architecture and training-data details.

| **Category** | **Model** | **Size** | **Layers** | **Attention heads** | **Training size** |
|---|---|---|---|---|---|
| Diffusion | LLaDA-Instruct | 8B | 32 | 32 | 2.3T |
| | Dream-Instruct | 7B | 32 | 32 | 580B |
| Autoregressive | Llama3 | 8B | 32 | 32 | 15T |
| | Qwen2.5 | 7B | 28 | 28 | 18T |

## D  ENCODING PERFORMANCE ACROSS LAYERS

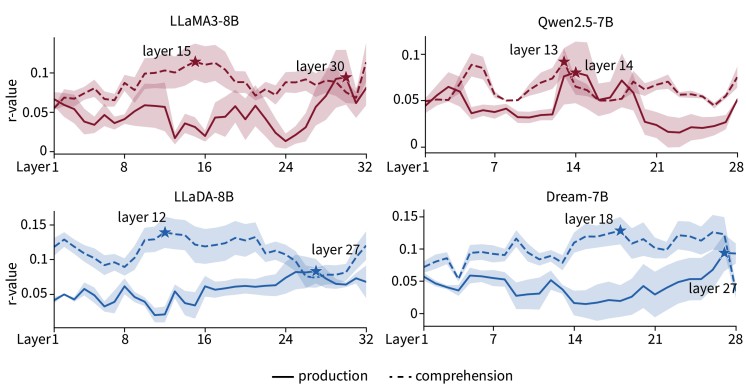

Figure 5: Encoding performance across layers for aLLMs (red) and dLLMs (blue). Solid lines represent production, whereas dashed lines represent comprehension.

## E  PROMPT EXAMPLES

**Conversation:**
Nurse (comprehension): `"Oh my hands are cold."`
Patient (production): `"Oh they do actually."`

**Prompt to LLMs:**
`In a casual conversation, you heard "Oh my hands are cold." and you responded "Oh they do actually."`

**Conversation:**
Nurse (comprehension): `"You're not gonna need it in two days. It's tem-porary."`
Patient (production): `"I know, but I feel like in two days they're proba-bly gonna go take this thing out of my head."`

**Prompt to LLMs:**
`In a casual conversation, you heard "You're not gonna need it in two days. It's temporary." and you responded "I know, but I feel like in two days they're probably gonna go take this thing out of my head."`

## F  CORRELATION BETWEEN AVERAGED AND LAST-TOKEN EMBEDDINGS

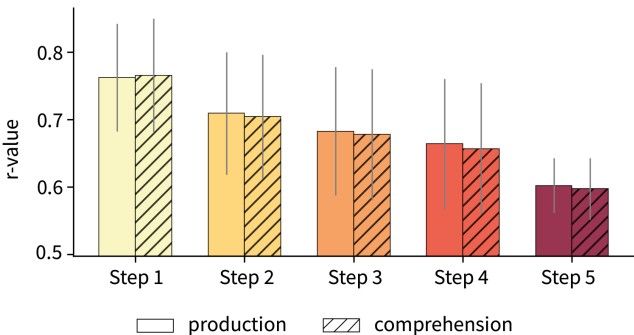

Figure 6: Step-wise correlation between averaged embeddings and last-token embeddings.

## G   BANDED RIDGE REGRESSION

Neural responses $\mathbf{y}$ were predicted as $\hat{\mathbf{y}} = \sum_i K_i w_i$, where $K_i = X_i X_i^\top$ and $w_i$ are kernel weights. The regression minimized $\|\mathbf{y} - \sum_i K_i w_i\|^2 + \sum_i \alpha_i w_i^\top K_i w_i$, with independent ridge penalties $\alpha_i$ per kernel. We used the `precomputed` kernel option with random search over $\alpha_i \in [10^0, 10^{20}]$, optimizing log-weights $\delta_i = -\log \alpha_i$ via cross-validation. Data were split 90%/10% into training and testing sets in temporal order. Per-kernel predictions $\hat{\mathbf{y}}_i$ were obtained using `predict(split=True)`, and Pearson correlations with observed responses were computed as model-specific scores. This was repeated for each embedding step (1–5) and timepoint, producing a tensor of size ($5 \times 15 \times 4$ models).

Statistical significance was assessed with non-parametric permutation tests. Correlation scores were aggregated across all electrodes, with LLaDA+Dream summed as diffusion and LLaMA+Qwen summed as autoregressive. Null distributions were generated from 1000 random permutations across 50,625 comparisons (5 steps $\times$ n electrodes $\times$ 15 timepoints), and $p$-values were computed as the proportion of permuted values exceeding the observed score.

## H   POSITION SHIFT OF WORDS IN DLLM ORDER

Table 3: Examples of original sentences and reordered outputs from LLaDA and Dream across five generation steps.

| Source | Step 1 | Step 2 | Step 3 | Step 4 | Step 5 |
|---|---|---|---|---|---|
| Original | Uh where | are they | going | for | dinner |
| LLaDA | dinner Uh | where they | are | going | for |
| Dream | dinner Uh | where they | are | going | for |
| Original | Mhm I | see my | sister and | her | husband |
| LLaDA | her husband | Mhm see | I my | sister | and |
| Dream | her husband | Mhm see | sister I | my | and |
| Original | Yeah Eh | it'll give | her | some | cushion |
| LLaDA | Eh cushion | Yeah it'll | give | her | some |
| Dream | her cushion | Yeah it'll | Eh | give | some |
| Original | We'll | try | our | best | together |
| LLaDA | try | We'll | our | best | together |
| Dream | try | We'll | best | our | together |
| Original | Not a | big soda | I don't | drink | soda |
| LLaDA | soda I | drink soda | Not big | a | don't |
| Dream | big soda | drink soda | Not I | a | don't |

We quantified the positional differences between the positions of words from dLLMs and their original positions in the sentence. The results show that words appearing early in the sentence (original Step 1) are unlikely to be placed in the final step (Step 5), and words originally occurring at the end of the sentence are more likely to appear in the later steps. However, words from the first step are distributed relatively evenly across Steps 2–4, indicating flexibility in the intermediate denoising stages.

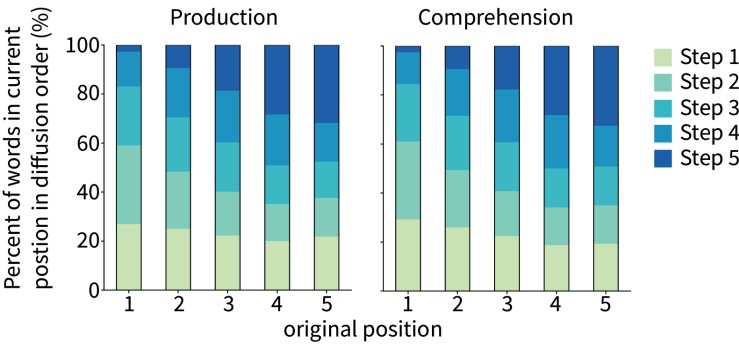

Figure 7: The x-axis indicates the original step in which a word appears, and the y-axis shows the percentage of words from each original position that appear in each dLLM step.

# I VARIANCE PARTITIONING

Table 4: Shared and unique variance explained by aLLM and dLLM across ROIs at Step 1

| ROIs | Comprehension (%) | | | Production (%) | | |
|---|---|---|---|---|---|---|
| | Shared | aLLM | dLLM | Shared) | aLLM | dLLM |
| STG | 25.28 | 42.67 | 32.05 | 24.60 | 34.89 | 40.51 |
| MTL+ITL | 17.30 | 39.67 | 43.03 | 5.73 | 50.06 | 44.21 |
| IFG | 28.17 | 40.16 | 31.66 | 14.41 | 52.39 | 33.20 |
| DLPFC | 8.98 | 61.62 | 29.40 | 29.04 | 28.22 | 42.74 |
| MC | 17.75 | 51.66 | 30.59 | 16.61 | 51.92 | 31.47 |
| AG | 28.41 | 36.73 | 34.86 | 23.14 | 39.07 | 37.79 |

Table 5: Shared and unique variance explained by aLLM and dLLM across ROIs at Step 2

| ROIs | Comprehension (%) | | | Production (%) | | |
|---|---|---|---|---|---|---|
| | Shared | aLLM | dLLM | Shared) | aLLM | dLLM |
| STG | 30.22 | 33.96 | 35.83 | 33.05 | 46.08 | 20.87 |
| MTL+ITL | 4.46 | 47.09 | 48.45 | 34.03 | 45.35 | 20.62 |
| IFG | 1.10 | 53.83 | 45.07 | 28.41 | 44.61 | 26.98 |
| DLPFC | 3.58 | 52.86 | 43.56 | 29.40 | 31.74 | 38.86 |
| MC | 16.04 | 55.35 | 28.61 | 26.95 | 49.42 | 23.64 |
| AG | 18.55 | 41.52 | 39.94 | 15.25 | 44.92 | 39.83 |

Table 6: Shared and unique variance explained by aLLM and dLLM across ROIs at Step 3

| ROIs | Comprehension (%) | | | Production (%) | | |
|---|---|---|---|---|---|---|
| | Shared | aLLM | dLLM | Shared) | aLLM | dLLM |
| STG | 21.46 | 42.66 | 35.88 | 22.57 | 33.88 | 43.55 |
| MTL+ITL | 25.56 | 40.18 | 34.26 | 23.63 | 22.91 | 53.46 |
| IFG | 13.31 | 45.70 | 40.99 | 2.52 | 44.33 | 53.14 |
| DLPFC | 7.82 | 47.45 | 44.72 | 33.68 | 32.62 | 33.70 |
| MC | 31.47 | 32.44 | 36.09 | 24.12 | 34.99 | 40.88 |
| AG | 12.22 | 44.87 | 42.91 | 2.05 | 48.75 | 49.20 |

Table 7: Shared and unique variance explained by aLLM and dLLM across ROIs at Step 4

| ROIs | Comprehension (%) | | | Production (%) | | |
|---|---|---|---|---|---|---|
| | **Shared** | **aLLM** | **dLLM** | **Shared)** | **aLLM** | **dLLM** |
| STG | 3.82 | 47.49 | 48.69 | 29.96 | 27.80 | 42.25 |
| MTL+ITL | 12.16 | 44.50 | 43.34 | 31.22 | 18.64 | 50.14 |
| IFG | 26.91 | 31.49 | 41.61 | 19.13 | 25.95 | 54.92 |
| DLPFC | 17.92 | 38.23 | 43.84 | 35.47 | 18.65 | 45.88 |
| MC | 25.13 | 36.98 | 37.88 | 1.15 | 34.19 | 64.66 |
| AG | 16.90 | 43.58 | 39.52 | 42.46 | 18.56 | 38.99 |

Table 8: Shared and unique variance explained by aLLM and dLLM across ROIs at Step 5

| ROIs | Comprehension (%) | | | Production (%) | | |
|---|---|---|---|---|---|---|
| | **Shared** | **aLLM** | **dLLM** | **Shared)** | **aLLM** | **dLLM** |
| STG | 19.68 | 39.99 | 40.33 | 43.96 | 24.86 | 31.18 |
| MTL+ITL | 4.03 | 50.96 | 45.01 | 57.41 | 17.77 | 24.82 |
| IFG | 22.06 | 34.74 | 43.20 | 39.80 | 24.32 | 35.88 |
| DLPFC | 29.17 | 30.06 | 40.77 | 47.87 | 17.57 | 34.57 |
| MC | 25.55 | 37.02 | 37.42 | 35.60 | 26.23 | 38.17 |
| AG | 16.31 | 41.89 | 41.80 | 50.67 | 15.00 | 34.33 |

# J ENCODING PERFORMANCE FOR ACOUSTIC AND SPEECH FEATURES

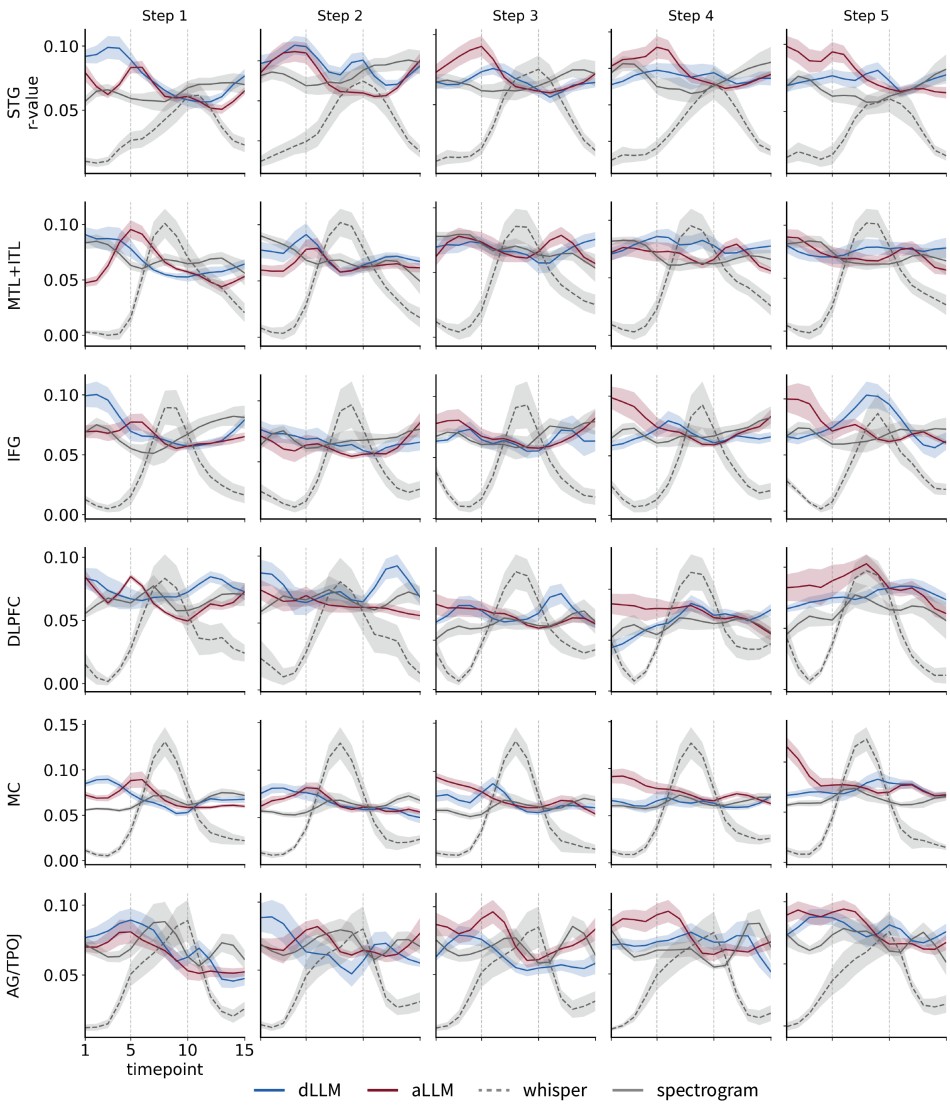

Figure 8: Encoding performance for aLLM, dLLM, acoustic and speech features during speech production. Sentence length is 5-25 words.

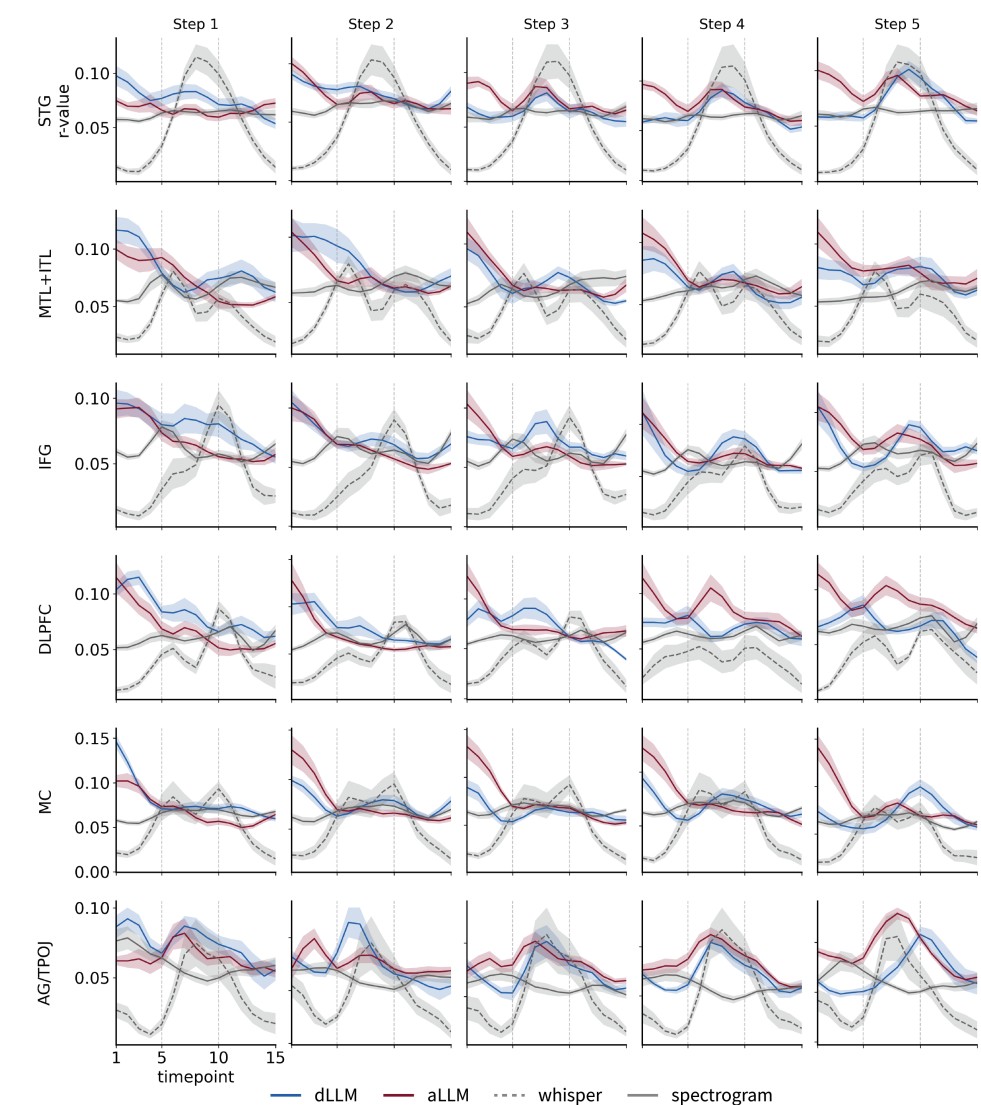

Figure 9: Encoding performance for aLLM, dLLM, acoustic and speech features during speech comprehension. Sentence length is 5-25 words.

## K  RANDOM CONTROL

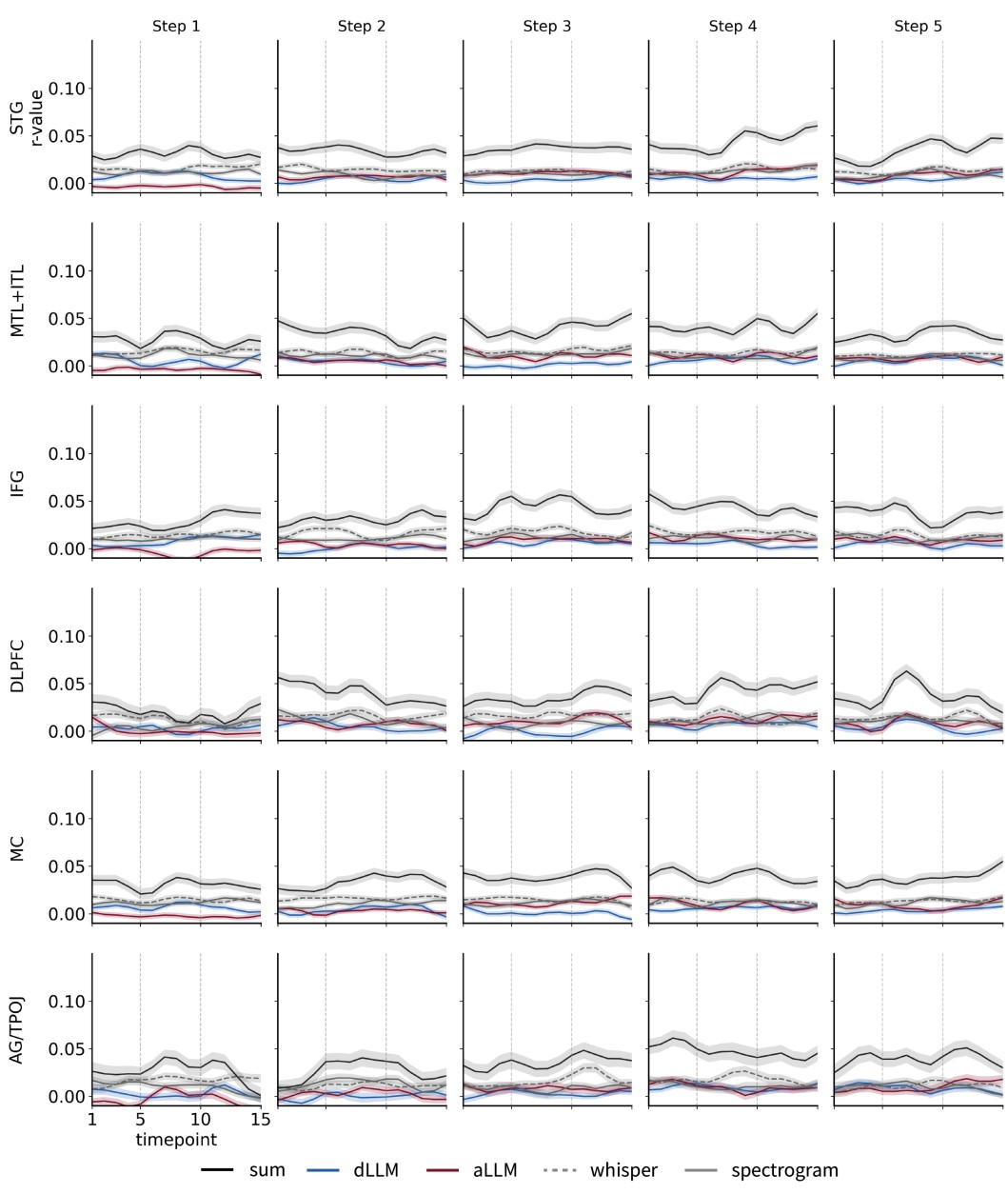

Figure 10: Encoding results for production sentences (5-25 word) with permuted ECoG data. No significant correlation was observed for all features.

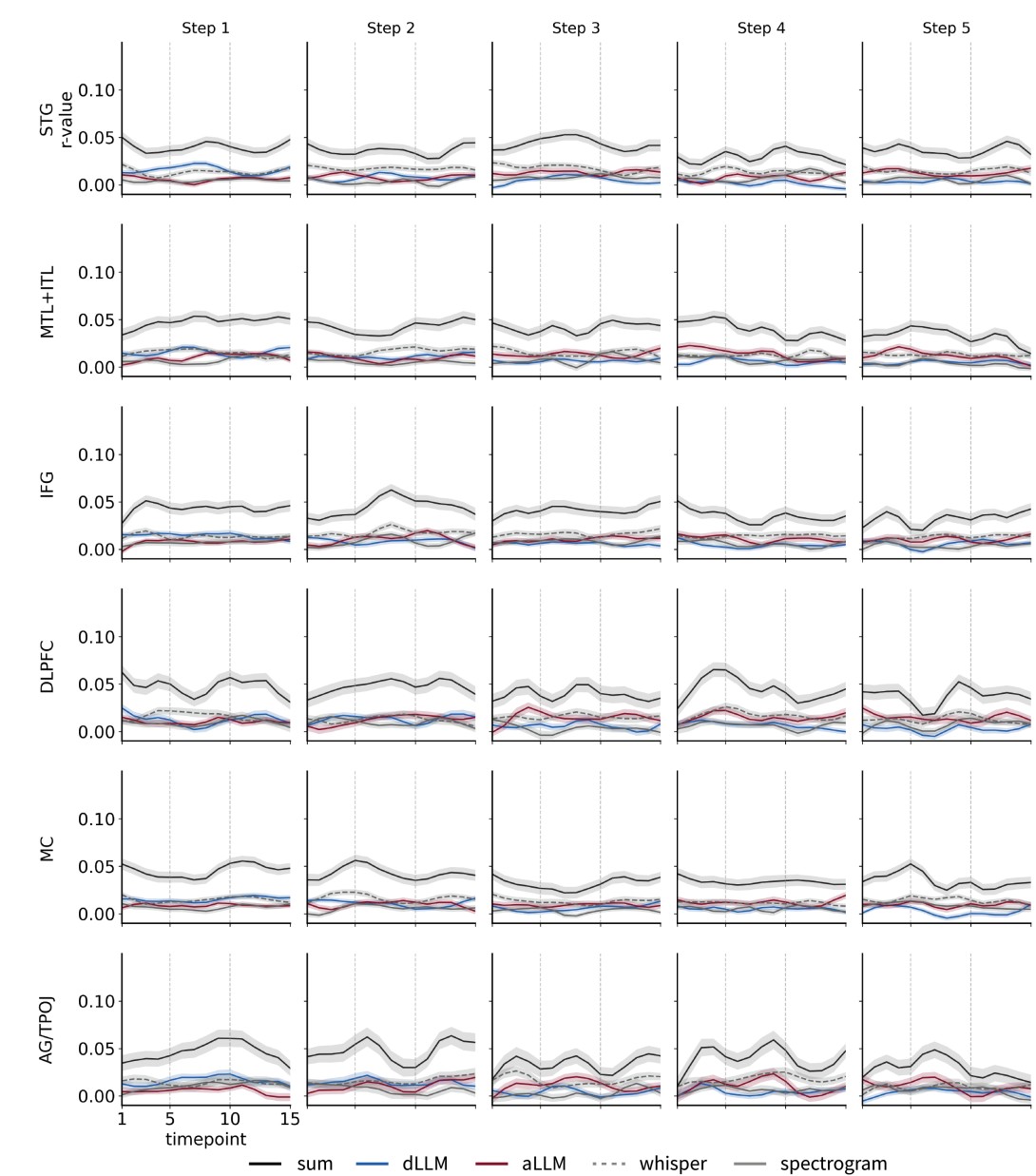

Figure 11: Encoding results for comprehension sentences (5-25 word) with permuted ECoG data. No significant correlation was observed for all features.

**a** Timecourse of encoding performance for permuted diffusion LLMs across steps in different brain regions for production

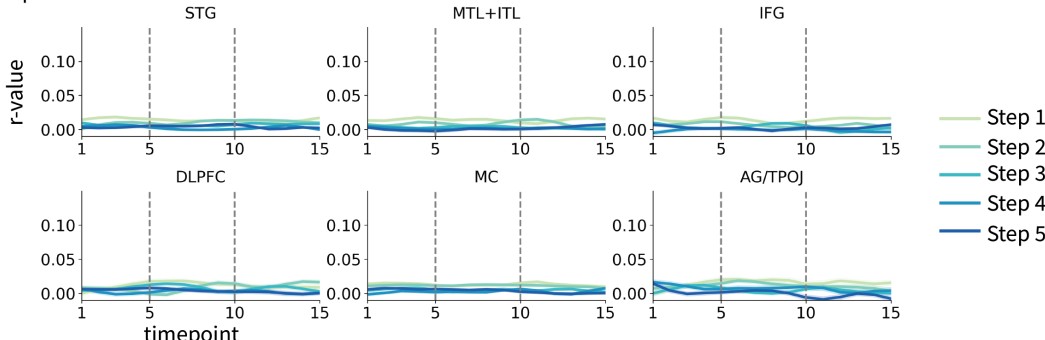

**b** Timecourse of encoding performance for permuted diffusion LLMs across steps in different brain regions for comprehension

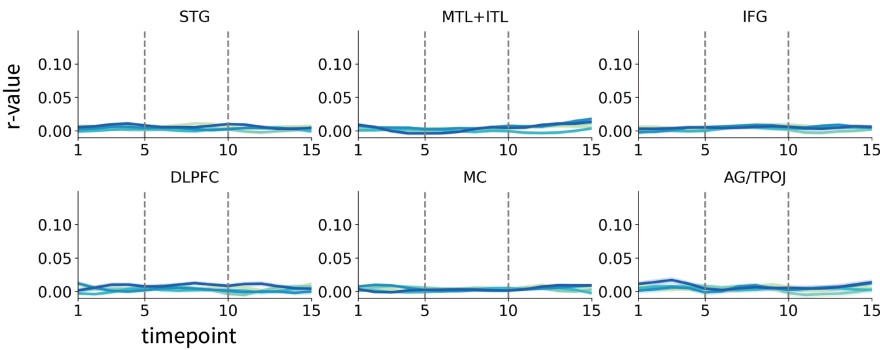

Figure 12: Encoding results for production sentences (5-25 word) with random relevation order of words in each step for dLLMs. No significant correlation was observed for all step embeddings.

## L    BRAIN ENCODING PERFORMANCE FOR LONGER SENTENCES

**a.** Timecourse of encoding performance for autoregressive LLMs across 10 steps in different brain regions for production

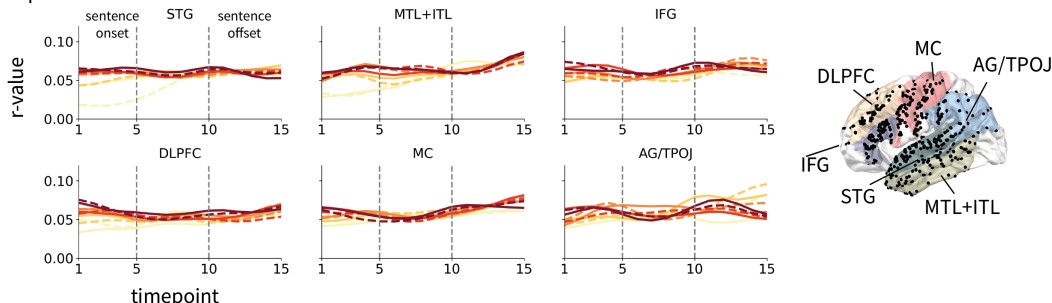

**b.** Timecourse of encoding performance for diffusion LLMs across 10 steps in different brain regions for production

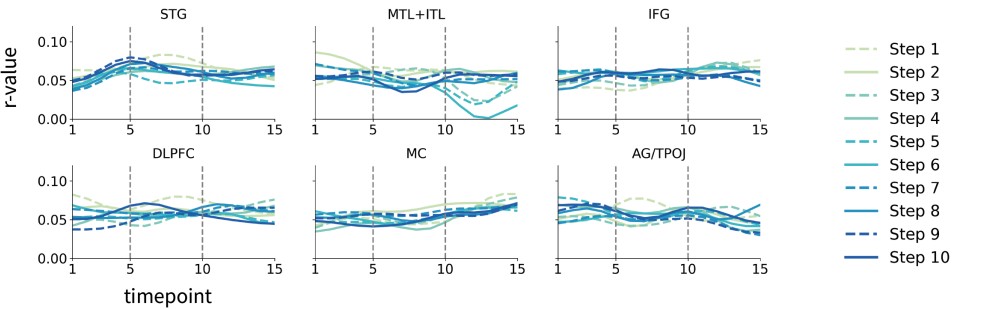

**c.** Timecourse of encoding performance for autoregressive LLMs across 10 steps in different brain regions for comprehension

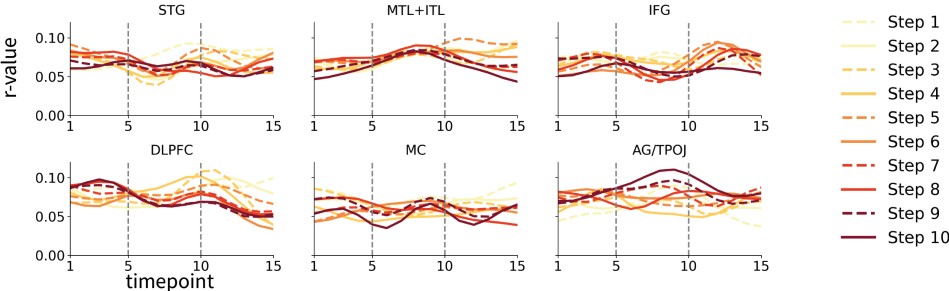

**d.** Timecourse of encoding performance for diffusion LLMs across 10 steps in different brain regions for comprehension

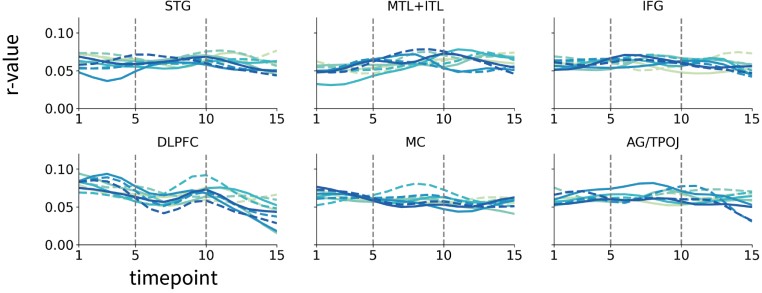

Figure 13: Encoding performance of aLLMs and dLLMs during **a,b** speech production and **c,d** comprehension. Sentence length is between 25-50 words.

# M   ENCODING PERFORMANCE OF FMRI DATA

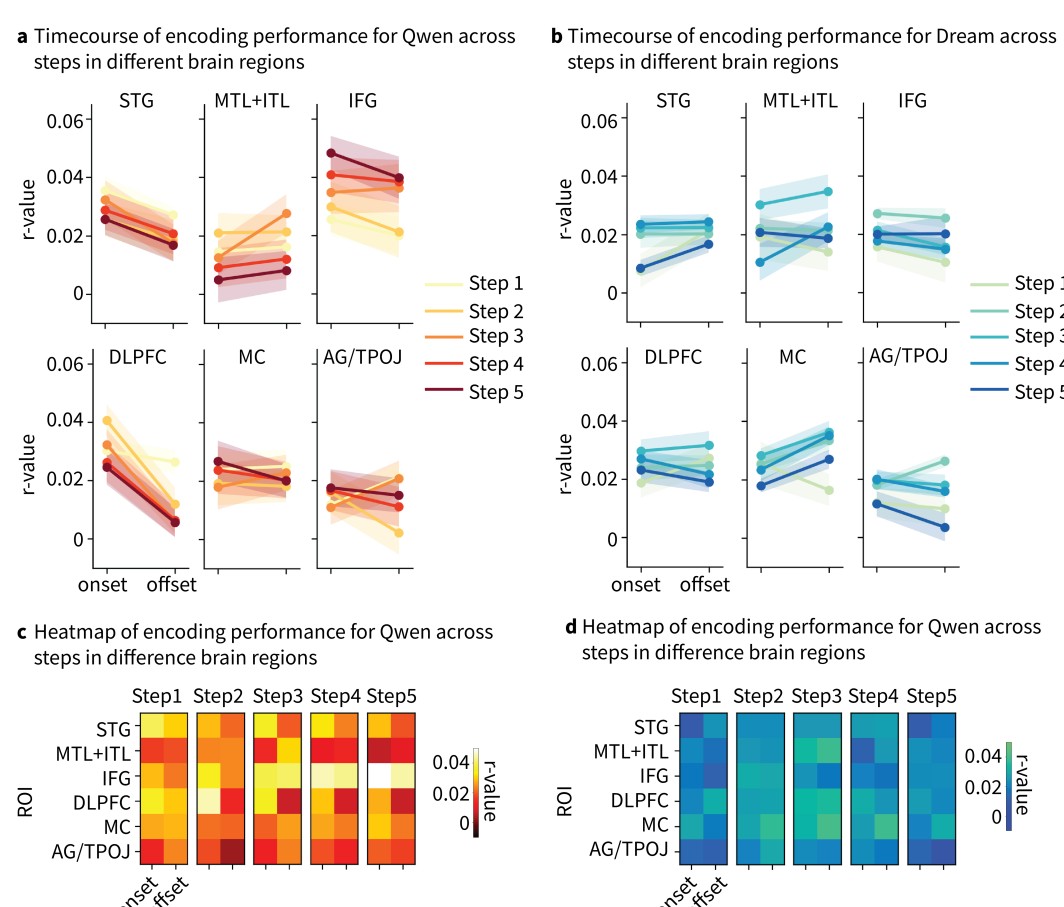

Figure 14: Brain encoding performance of fMRI data during speech comprehension.

# N   ENCODING PERFORMANCE USING LAST-TOKEN EMBEDDINGS OF ALLMS

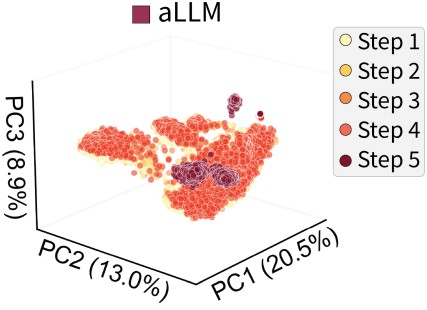

Figure 15: 5-step last-token embeddings for aLLM after PCA.

**a.** Timecourse of encoding performance for autoregressive LLMs across all steps in different brain regions for production

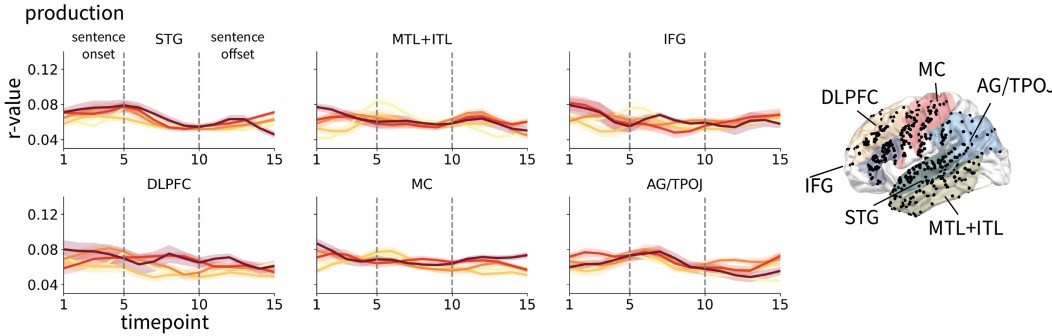

**b.** Timecourse of encoding performance for diffusion LLMs across all steps in different brain regions for production

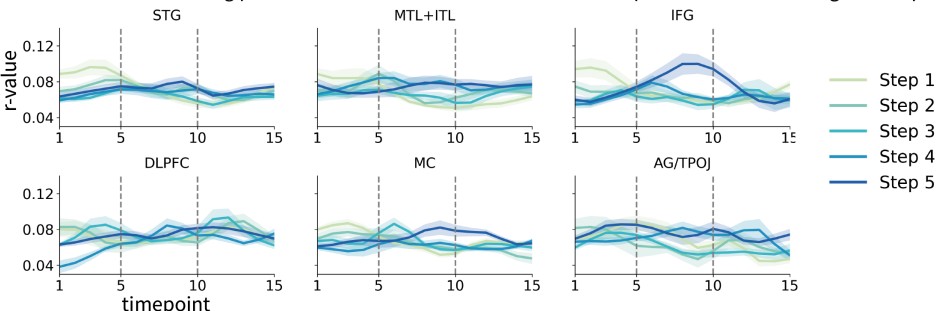

**c.** Timecourse of encoding performance for autoregressive LLMs across all steps in different brain regions for comprehension

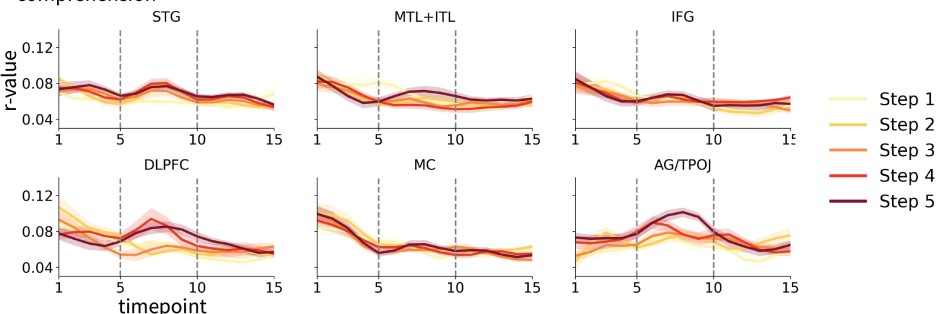

**d.** Timecourse of encoding performance for diffusion LLMs across all steps in different brain regions for comprehension

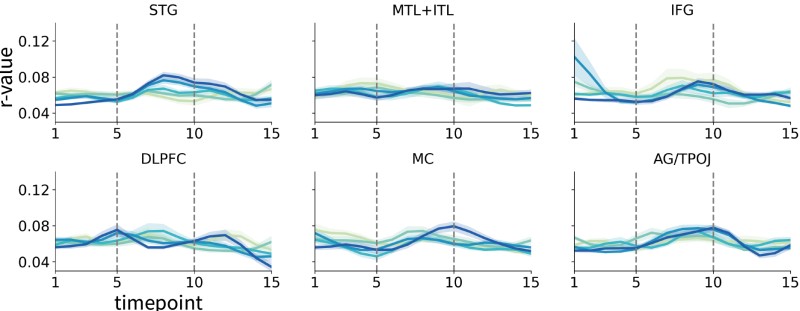

Figure 16: Brain encoding with last-token embeddinsg from aLLMs at each step and averaged dLLM embeddings at each step.

## O   SENTENCE DATA

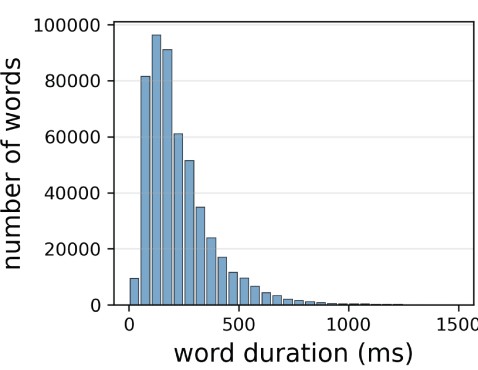

Figure 17: Summary statistics of all sentences.

## P   IMPLEMENTATION DETAILS

All computations for extracting dLLM and aLLM sentence embeddings and for running ridge regression brain-encoding analyses were performed on a high-performance computing (HPC) cluster equipped with 128 CPU cores and two A100 GPUs per node.

