# OpenReview forum: "Diffusion-based dynamics as a cognitive model of human speech production"
_ICLR.cc/2026/Conference — Submitted to ICLR 2026_

### Official Review · Reviewer_UqtJ · 2025-10-27

**Soundness:** 2
**Presentation:** 3
**Contribution:** 4
**Rating:** 6
**Confidence:** 4

**Summary:**

This paper tests whether diffusion-based large language models (dLLMs) better align with human brain activity during speech production than traditional autoregressive LLMs (aLLMs), which generate text sequentially. dLLMs iteratively refine entire sentence representation. This process may more closely mirror how the human brain processes linguistic information.

The study uses electrocorticography (ECoG) recordings from four patients producing natural speech. The authors compare intermediate denoising steps of dLLMs (LLaDA-8B and Dream-7B) with their autoregressive counterparts (LLaMA3-8B and Qwen2.5-7B).

They find that dLLM embeddings explain more neural variance during both pre- and post-articulatory phases, particularly in middle and inferior temporal as well as motor regions. Early diffusion steps correlate with pre-articulatory activity, while later steps align with articulation and monitoring processes.

Overall, their findings support iterative refinement as a plausible mechanism for speech planning.

**Strengths:**

- Quality of writing: The paper is very well written. It clearly states its contributions and situates them in the literature with precision (explaining what dLLMs are, context of psycholinguistics, model–brain alignment). The figures are clear and informative.

- Novel hypothesis and framing: The idea that diffusion dynamics could map onto human speech planning is original and makes sense in the context of model-brain alignment. It challenges the dominance of autoregressive models in brain–language alignment. To my knowledge, this is the first paper attempting such an experiment.

- Data quality: The use of naturalistic ECoG speech production data (4 participants, 50 hours) is extremely valuable. The temporal precision of ECoG allows for fine-grained mapping of pre- and post-articulatory processes.

- Careful model design: The authors implement a psychologically interpretable “greedy confidence” revelation algorithm for diffusion steps and ensure comparability by matching autoregressive steps within the same model. Autoregressive versions of the same architectures are used as controls.

- Comprehensive analysis and discussion: The study includes both lexical-level statistics (frequency, POS) and representational-level analyses (PCA trajectories, JS divergence, encoding correlations). The authors remain cautious in their interpretation, clearly noting that their results are based on a high-quality but limited dataset and emphasizing the need for replication with other modalities such as fMRI and MEG.

**Weaknesses:**

- Data quality: What is a strength is also a limitation. The dataset includes only four participants. While the data are of exceptional quality, this raises concerns about generalizability to other modalities such as MEG or fMRI. The small sample size also limits the statistical rigor of some analyses.

- Limited baselines: The study tests only two diffusion models and their autoregressive counterparts at a medium scale (7–8B parameters). Results might differ with larger or multimodal models, which makes the conclusions somewhat model-specific.

- Potential confound: The dLLM and aLLM comparison involves algorithmic differences: confidence-based token revelation versus prefix growth. Neural alignment differences may therefore reflect these distinct operational mechanisms rather than the diffusion process itself.

- Fixed analysis parameters: The authors use a fixed layer (layer 20) and a fixed number of diffusion samples (5) for their analyses. It would be important to verify that these choices are indeed optimal or robust across settings.

- Minor issue: Typo at line 782: “Brian Encoding.”

**Questions:**

1. Could the authors test whether their results generalize beyond ECoG? For instance, by evaluating model–brain alignment using fMRI or MEG data with publicly available datasets for comparison?

2. How sensitive are the results to the choice of diffusion layer (layer 20) and the number of denoising samples (5)? Would varying these parameters change the alignment patterns observed?

3. To what extent could the observed alignment differences between dLLMs and aLLMs be explained by the algorithmic disparity between confidence-based revelation and prefix growth? It's unclear whether it changes entirely the way of analyzing the results.  Would random or frequency-based token revelation produce similar neural alignment patterns? This would test if the greedy-confidence rule is critical.

4. Have the authors explored whether larger or multimodal diffusion models reproduce the same alignment advantage, or whether this effect is specific to mid-sized architectures (7–8B)?

5. Given the limited number of participants, can the authors provide effect-size estimates or subject-level analyses to assess the robustness and inter-individual consistency of their findings? How stable are the diffusion–brain correlations across the four patients?

---

> ### Author Response · Authors · 2025-11-25
>
> Thank you very much for your constructive feedback, which has greatly helped us improve the paper. Below are our point-by-point responses to the weaknesses and questions.
>
> **Weakness1**
>
> RE32: Although our dataset includes only four patients, each participant was recorded continuously over multiple days (24/7), yielding approximately 100 hours of neural data in total. This level of within-subject sampling is not comparable to typical fMRI datasets, which usually contain only 1–2 hours of data per subject. We therefore believe that the dataset provides sufficient statistical power. Indeed, many prior studies have included only 3–4 participants but with extensive neural data per individual (e.g., Goldstein et al., 2022, 2024, 2025; Huth et al., 2016; Tang et al., 2023). Another reason we did not use large open-access fMRI datasets with hundreds of participants is that diffusion-based mechanisms are most relevant to language production, given that diffusion models are generative. In contrast, language comprehension proceeds incrementally as words are heard, and is therefore more naturally aligned with autoregressive processing. Current open fMRI/MEG datasets are predominantly focused on comprehension tasks, as language production typically induces head movement that can compromise data quality. That said, as Reviewer UqtJ also suggested, we have now included results from an open fMRI dataset of naturalistic listening consisting of approximately two hours of story comprehension data from 49 English participants (Li et al., 2022). We observed overall higher alignment for last-step aLLM embeddings in IFG and for mid-step dLLM embeddings in MTL/ITL, but no consistent or interpretable temporal pattern emerged. These results further suggest that the diffusion-based representations are most relevant for production, whereas comprehension does not exhibit the same diffusion–brain alignment (see Figure 14 in Appendix M).
>
> **Weakness2**
>
> RE33: We agree that model size can influence encoding performance. However, to our knowledge, the 7B/8B versions used here are currently the largest openly available diffusion-based LLMs.
>
> **Weakness3**
>
> RE34: We understand the concern, but note that the confidence-based token revelation procedure is an integral part of how diffusion LLMs are trained and how they perform generation. As such, it is not an arbitrary design choice but a core operational component of the diffusion process itself. Therefore, differences in neural alignment that arise from confidence-based revelation versus prefix growth can reasonably be interpreted as reflecting the underlying generative mechanisms of dLLMs rather than an external confound.
>
> **Weakness4**
>
> RE35: We have now performed ridge regression analyses across all layers of the four LLMs. For comprehension sentences, the best-performing layers for LLaMA, LLaDA, Qwen, and Dream were Layers 15, 12, 13, and 18, respectively. For production sentences, the corresponding best layers were 30, 27, 14, and 27 (see Figure 5 in Appendix D). All results in the revised manuscript have been recomputed using these best-layer embeddings for each LLM.
>
> **Typo**
>
> RE36: We have now removed the individual results from the two aLLMs and dLLMs which contains the typo.

---

> > ### Author Response · Authors · 2025-11-25
> >
> > **Question1**
> >
> > RE37: Current open fMRI/MEG datasets are predominantly focused on comprehension tasks, as language production typically induces head movement that can compromise data quality. However, we are more interested in language production as dLLMs is mainly a generation process. We have still included encoding results using the same analyses with an open fMRI dataset where 49 English speakers listen to a 100-minute long audiobook in the scanner (Li et al., 2022). Our results showed higher alignment for last-step aLLM embeddings in IFG and for mid-step dLLM embeddings in MTL/ITL, but no consistent or interpretable temporal pattern emerged, suggesting that the diffusion-based representations are most relevant for production (see Figure 14 in Appendix M).
> >
> > **Question2**
> >
> > RE38: We have now performed ridge regression analyses across all layers of the four LLMs. For comprehension sentences, the best-performing layers for LLaMA, LLaDA, Qwen, and Dream were Layers 15, 12, 13, and 18, respectively. For production sentences, the corresponding best layers were 30, 27, 14, and 27 (see Appendix Figure). All results in the revised manuscript have been recomputed using these best-layer embeddings for each LLM.
> >
> > **Question3**
> >
> > RE39: The confidence-based token revelation procedure is an integral part of how diffusion LLMs are trained and how they generate text. It is therefore not an arbitrary design choice but a core operational component of the diffusion process itself. To further address this concern, we conducted the same analyses using a random token-revelation order. We did not observe any alignment patterns in any ROI under this control condition (see Figure 12 in Appendix K), indicating that the alignment observed with confidence-based revelation is not arbitrary and indeed reflects the intended generative mechanism of dLLMs.
> >
> > **Question4**
> >
> > RE40: To the best of our knowledge, the 7B/8B versions used here are currently the largest openly available diffusion-based LLMs.
> >
> > **Question5**
> >
> > RE41: We have now added standard-error shading to all relevant plots to illustrate variability across participants. As shown, the diffusion–brain correlations are relatively consistent across the four patients, with limited inter-individual variance.

---

> ### Comment · Reviewer_UqtJ · 2025-11-26
>
> Thank you for your answers ! Several of your responses make it clear that I had misinterpreted parts of the methodology. I feel that you have added information that enables better clarity than in the original version. Thank you for your explanations and clarifications especially the addition of the fMRI control.
>
> Given this, I will keep my overall evaluation unchanged as I am still convinced this is a valuable work but I will downgrade my confidence score from 4 to 3.

---

> > ### Author Response · Authors · 2025-11-27
> >
> > Thank you very much for acknowledging the effort we put into our responses. However, lowering the confidence score effectively reduces the overall evaluation. If you feel that our revisions and clarifications have improved the quality of the work, we would greatly appreciate it if you could reflect that in the evaluation.

---

### Official Review · Reviewer_Gw42 · 2025-10-27

**Soundness:** 2
**Presentation:** 3
**Contribution:** 2
**Rating:** 4
**Confidence:** 5

**Summary:**

This work is part of a broader research effort to explore the relationship between internal representations of large language models (LLMs) and human brain activity recorded during speech production. Specifically, the authors focus on comparing diffusion LLMs (dLLMs) (construct sentences iteratively) with autoregressive LLMs (aLLMs) (follow next-word prediction with a left-to-right sequence) to examine which LLMs better mirrors human-like speech production. The evaluation focuses on comparing the sentence representations across steps in two types of LLMs and demonstrates how the embeddings extracted from the stepwise representations of the target sentences are used in a neural encoding model for predicting ECoG brain activity in speech production. The authors compare brain encoding performance across two types of LLMs to examine neural dynamics in both language and motor regions. Overall, diffusion LLMs provide a new way to iteratively generate sentences and reorganize lexical categories across the course of generation differently than autoregressive LLMs.

**Contributions:**

* *Alignment between diffusion LLMs and ECoG activity during naturalistic speech prediction:* The study extract the internal embeddings from diffusion LLMs across denoising steps and use these representations in neural encoding model to predict ECoG brain activity during speech production, which is methodologically novel. This framing provides a new comparison for language production, whereas autoregressive LLMs are frequently used in language comprehension.
* *Comprehensive evaluation:* The study compares diffusion LLMs and their autoregressive versions, assessing iteratively denoising global representations from dLLMs and left-to-right sequence representations from aLLMs and relates these representations with brain recordings during speech production. For each encoding model, the authors measure the correlation between actual and predicted brain activity. Further, they compare encoding performance across both types of LLMs, and quantify correlation performance across generative steps in sentence construction and six cortical ROIs.

**Technical summary:**
This is primarily an empirical study, and its methodology involves the following components:
* *Embeddings from diffusion and autoregressive LLMs:* The authors use both diffusion LLMs and autoregressive LLMs to extract the embeddings during speech production. For the autoregressive approach, they follow standard left-to-right generation with five progressive prefixes (20-100%) and extract sentence embeddings at each step. For diffusion LLMs, an iterative denoising approach reveals the best possible word at any position in the sentence; fix the revealed words, and next iterations fill the remaining masks to complete the sentence. Embeddings from each iteration step are then used for encoding.
* *Neural encoding model:* To train neural encoding model, the authors use banded ridge model, where multiple embedding sources can be combined and passed as input to predict target ECoG recordings. The model is minimised using MSE; ridge parameters are independent per kernel and selected via randomized search over the precomputed kernels. Using the split=true option provides each embedding specific predictions.
* *ECoG conversation dataset:* The authors use an available ECoG dataset, where four patients engaged in natural conversation with family, friends, and doctors, yielding 50 hours of comprehension and 50 hours of production data. The authors use production data, considering utterance between 5-25 words in length.

**Experimental design/evaluation:**

* *Word-generation dynamics across diffusion steps:* The authors evaluate the word generation across diffusion steps in diffusion LLMs and how these temporal trajectory changes differ from autoregressive LLMs. To quantify the distributional differences, they compute Jensen–Shannon divergence between model families across which shows how generation happens between steps.
* *Neural encoding performance:* This analysis evaluates whether the step-wise representations extracted from diffusion LLMs better predict ECoG than from autoregressive LLMs. The primary questions are (i) how the encoding performance varies across five generative steps and six cortical ROIs, and further check if there are any differences in activation patterns for early vs. late diffusion steps.

**Main findings:**
According to the authors’ interpretation, the main findings are as follows:
* dLLMs (25.2%) show stronger step-wise differentiations than aLLMs (9.7%).
* dLLMs rely mostly on high-frequency words at sentence boundaries (especially at start of sentence) and low-frequency words at mid-sentence, whereas aLLMs show opposite tendency at middle steps.
* In dLLMs, early diffusion steps are strongly correlated with temporal regions (STG, MTL+ITL), while later diffusion steps show high correlation in STG, IFG, AG.

**Strengths:**

I found this work to have the following strengths:
* *Clarity:* The manuscript is well written and well structured. The pipeline in Figure 2 is easy to follow, and clearly presents the differences in generation steps between dLLMs and aLLMs. Later, the greedy-based token revelation algorithm is described clearly.  The banded (multi-kernel) ridge encoding model is explained well, including how model-specific predictions are obtained. The results section clearly reports step-wise differences in neural encoding across the two model families and relates denoising-step representations to activity across cortical ROIs as step index increases.
* *Originality:* The idea of using diffusion LLMs during speech production in a neural encoding model to predict ECoG is a simple but methodologically novel contribution.. Prior brain encoding studies typically use autoregressive LLMs for both comprehension or production and learn a ridge-regression model. In contrast, dLLMs provide global, iterative refinement of the whole sentence, whereas aLLMs generate left-to-right based only on preceding tokens, offering a complementary approach for pre-articulatory planning.
* *Significance:* This work is significant in that it contributes to a better understanding of the parallels between language-	 models dynamics and language processing in the human brain over time during speech production. It shows that diffusion LLMs capture neural dynamics qualitatively differ with autoregressive LLMs, and suggest that diffusion LLMs offers global approach that might be a better parallel for how human plan utterances than left-to-right generation in autoregressive LLMs.

**Weaknesses:**

From my perspective, the primary weaknesses of this study arise from the lack of comparison with prior literature, and limited evaluation:
* *Limited model evaluation:*
    * The prior work on ECoG data considered speech-to-text language model (Whisper) and  the autoregressive LLM (GPT-2) to perform brain encoding on both speech comprehension and speech production [Goldstein et al. 2025]. In particular, they show that the encoding model accurately predicts neural activity at each level of the language processing in both comprehension and production. However, the current study focuses only on speech production and considers only text-based representations from dLLMs, concluding that dLLMs outperform aLLMs on production.
    * Speech production relies on both acoustic and auditory-pathways alongside language production. However, this study is limited to text representations across denoising steps. which misses the low-level speech dynamics that are crucial for motor/sensory regions. As a result, the claim that the diffusion LLMs encoder better than autoregressive LLMs for production is underdetermined and may reflect missing pathways rather than diffusion

* *Small sample size and low temporal resolution:* The most significant limitation of current study is that they restrict  utterances to 5–25 words words in length by not considering both short and long utterances. This may be the fact that the current study reports very low encoding performance for both diffusion LLMs ($\sim$0.05) and autoregressive LLMs ($\sim$0.03) across six cortical ROIs. In particular, the performance of autoregressive LLMs is significantly lower than prior work that analyzes full comprehension and production datasets and reports higher correlations ($\sim$r > 0.15, Goldstein et al., 2025) across language and motor regions. As a result, the claims made in the paper are undermined by limited dataset and temporal resolution rather than model choice is unclear.
     * Sampling ECoG activity at evenly spaced bins is not ideal, as prior study considered full comprehension and production dataset using 25 ms windows from −2000 to +2000 ms around word onset. Ignoring full ECoG activity by considering only evenly spaced bins may limit language-related activity.
* *Lack of baseline results:* The paper compares only two categories of LLMs without reporting any baseline results. Without a strong baseline model, it is hard to make claims about whether these LLMs are strong at predicting ECoG and have better encoding performance. A strong baseline would establish what these  LLMs predict beyond simpler models, and prior work.
* *No standard error bars across subjects:* The results reported in Figure 4 are  mean values only for diffusion LLMs and autoregressive LLMs, without accompanying measures of variability such as standard error across subjects. These metrics are critical for evaluating the robustness of the findings, as this may explain each subject’s specific results in production.
For a complete and detailed account of both major and minor issues, please refer to the “Questions” section.

**Questions:**

I would like to thank the authors for the interesting comparison of dLLMs and aLLMs in ECoG encoding during speech production in this work. However, there are several points that I believe require further attention/work. I have divided these into major issues, which should be prioritized, and minor ones, which should be addressed for a strong version of current work.

**Major Comments/Questions:**
* *Small sample size:* While I fully understand the complexity associated with empirical research involving brain datasets, the sample size in this study appears to be very limited. Similar to Goldstein et al. 2025 study, I strongly encourage the authors to consider a full 50hours production ECoG dataset that includes both short and long utterances, and compare the findings of dLLMs and aLLMs with prior approaches. If aLLMs replicate prior performance to Goldstein et al. 2025] work, and dLLMs still result in superior encoding performance than aLLMs across six cortical ROIs, then the authors provide a clear justification for the claims made in the paper and its sufficiency for the conclusions drawn.
* *Comparison with baselines:* I recommend authors to compute strong baselines such as acoustic features (log-mel or MFCC), mid-level speech embeddings (Whisper encoder/decoder) as these are prior pathways for language production. Then compare two types of LLMs against baselines to establish (i) absolute performance and (ii) whether dLLMs/aLLMs explain unique variance beyond the baselines.
* *Variance partitioning on two types of LLMs:* Since authors used the banded ridge model to train encoding models across feature spaces, each kernel may learn unique explainable variance, and shared between them during speech production. I strongly recommend authors to perform following analysis
     * Perform variance partitioning on both dLLMs and aLLMs to analyze the unique and shared variance of these LLMs across six cortical ROIs.
     * Using aLLMs, Goldstein et al. 2025 report higher encoding performance in IFG during speech production. I recommend authors to compare this analysis with prior approaches.
     * I recommend authors to present a ROI × (Unique dLLM, Unique aLLM, Shared) table/heatmap.
     * This analysis can be conducted at each diffusion step as well.
* *Reporting variability:* Please report standard errors across four participants alongside mean values to provide a clearer picture of variability in Figure 4 for both dLLMs and aLLMs. Even with a small dataset, such measures are crucial for assessing the robustness and reliability of the results.
* *Overclaims and Scope:*  With limited sample size, constrained utterance lengths, only production data selection, the claim that dLLMs are more “cognitively plausible” or “human-like” than aLLMs is overstated. Further, the authors only considered five diffusion steps with a greedy-approach, target-aware revelation approach, while aLLMs are evaluated with fixed word distribution with left-to-right prefixes making these are directly comparable. Please soften the language and add matched comparisons (e.g., target-agnostic dLLM steps and/or a rightward-only, non-bidirectional mask-fill variant ) to isolate the effect of diffusion from bidirectionality.
* *Layer selection rationale:* The choice of layer 20 is justified by prior work on language comprehension, but those studies (e.g., Caucheteux et al., 2022) did not use LLaMA-2 or Qwen-2.5. Selecting “layer 20” across different architectures and tasks lacks grounding. The best layer selection should be based on prior work using the same architecture or, preferably, on empirical validation. Follow Antonello et al. 2023 for best layer selection for LLaMA-2. I recommended authors to empirically select the best layer for Qwen-2.5.

**Minor Comments/Typos:**
While addressing the following points may not be critical to the paper’s core contributions, doing so would enhance the overall quality.
* Figure 3: Please fix the Y-axis for Figure 3b and 3c.
* Figure 3: Clarify how the percentage measures are computed.
* Line 165: Across both types LLMs, are there any differences in training data? initialization, or architecture? Whether the underlying backbone and number of layers are the same across models? Please report a table with model training details, parameters, #layers.
* Please justify the choice of (steps =5 ) diffusion steps. Does increasing the number of steps change contextual representations or brain-prediction performance? Authors can provide examples by considering steps=10 (generating sentences by considering 10 denoising steps).
* Goldstein et al. (2025) show clear POS clusters in language embeddings, but current study using aLLMs step representations look cluttered with no obvious POS structure. Please clarify why. Concretely.

**General Advice:**
The manuscript presents a comparison study of two types of LLMs in ECoG encoding during speech production and a range of experimental design choices for stepwise feature extraction during denoising and building encoding models. However, the current version lacks a clear comparison with previous work that evaluates both comprehension and production for the same dataset, relies on small sample size and limited experimental evaluation. Adding explicit implications and addressing the above mentioned weaknesses and major comments would make the work stronger.

---

> ### Author Response · Authors · 2025-11-25
>
> We thank you very much for the highly constructive feedback. Below, we provide our point-by-point responses to the identified weaknesses and questions.
>
> **Weakness1**
>
> RE16: We agree with your concern. However, the main goal of this paper is to explore whether diffusion LLMs could explain some aspects of language production in the human brain, and our focus on text-based models reflects an important practical limitation: at present, there is no diffusion-based speech LLM available. We have added this limitation to the Discussion section. To address the role of acoustic and speech information, we also included the spectrogram and the last encoder layer of Whisper (averaged over all tokens in each sentence) as control variables in the banded ridge regression. The results showed that Whisper achieves good model fit in the middle-to-late time windows in the left STG, MTL/ITL, DLPFC, and MC. However, text-based aLLMs and dLLMs still explain additional variance beyond these auditory predictors. In particular, earlier-step dLLM embeddings continue to fit best in the early time windows of MTL/ITL during speech production, whereas aLLMs show peak performance across all timepoints in MTL/ITL (see Figure 8-9 in Appendix J).
>
> **Weakness2**
>
> RE17: Thank you for the suggestion. As explained in RE12 to Reviewer sif2, we selected the 5–25 word range because it reflects typical utterance lengths in everyday communication. Sentences approaching 25 words (e.g., “hard for me to swallow nine grams of protein the protein when you're not eating protein turns into carbohydrates plenty of it's plenty of cranberries”) already tend to include many repetitions, hesitations (e.g., “um”), or repairs, and extending far beyond this range (e.g., 25–50 words) often results in unusually long and less fluent utterances. However, we agree that sentence length may influence encoding performance and we have now added analyses using comprehension and production sentences between 25–50 words. The results show that earlier-step dLLM embeddings continue to better predict neural responses in MTL/ITL and motor cortex during production (see Figure 13 in Appendix L), consistent with the findings from the 5–25 word range.
>
> **Weakness3**
>
> RE18: Our rationale for sampling five steps per sentence stems from the fact that sentences vary in length. Unlike prior studies that focus on word-level alignment, our analysis centers on entire sentences and evaluates encoding performance across progressive stages of sentence formation. Sampling five evenly spaced steps allows us to standardize the number of timepoints per sentence while directly addressing our research question: whether earlier denoising steps of diffusion models align with earlier neural time windows, and whether later steps align more strongly with later time windows.
>
> **Weakness4**
>
> RE19: We have now added two baseline models: the mean spectrogram over the time period of each sentence, and the last layer of the encoder of Whisper averaged over all tokens in each sentence. Our current banded ridge regression model includes 6 sets of features: spectrogram, Whisper, LLaMA, LLaDA, Qwen and Dream. We summed the correlation coefficients of LLaMA and Qwen to represent unique variance explained by aLLMs, and summed the correlation coefficients of LLaDA and Dream to represent unique variance explained by dLLMs. Our results showed additional variance explained by our aLLMs and dLLMs above the baseline models (see Figure 8-9 in Appendix J).
>
> **Weakness5**
>
> RE20: We have now added shading to all line plots which represents one standard error. We first computed the mean over significant sensors within each ROI for each subject, we then computed the standard error at the group level.

---

> > ### Author Response · Authors · 2025-11-25
> >
> > **Question1**
> >
> > RE21: We have now added results from longer sentences (25–50 words) as well as from comprehension sentences. For these longer sentences, we also increased the number of denoising steps to 10. The results show that, in production, earlier-step dLLM embeddings continue to outperform aLLMs in the left MTL/ITL, consistent with the findings from shorter sentences (see Figure13a in Appendix L). This pattern, however, is not observed for comprehension sentences (see Figure13b in Appendix L).
> >
> > **Question2**
> >
> > RE22: We have now included 2 baseline regressors: (1) spectrogram averaged over the length of each step of sentence and (2) the last layer of Whisper encoder averaged over all tokens in each step of sentence. These features were entered into the banded ridge regression together with our aLLM and dLLM embeddings. For production, we observed higher alignment of Whisper embeddings in sentence-middle timepoint in the MTL/ITL and MC across all steps (see Figure 8 in Appendix J). For comprehension, Whisper showed highest correlation in the middle-sentence position in the STG. Spectrogram does not show better model fit compare to other regressors in both production and comprehension (see Figure 9 in Appendix J).
> >
> > **Question3**
> >
> > RE23：We have now performed variance partitioning on dLLMs and aLLMs across the six ROIs. Specifically, for each step, we first conducted 2 ridge regressions, first with the two aLLM (LLaMA and Qwen) embeddings, then with the two dLLM embeddings (LLaDA and Dream). We then conducted banded ridge regression with the 4 sets of embeddings. The shared variance is the difference between the correlation coefficients of the two ridge regressions and the banded ridge regression. We computed percentages of the shared variance and unique variance over the overall correlation coefficients of the banded ridge regression. The results are shown in Tables 4-8 in Appendix I.
> >
> > The prior results are based on word-level encoding performance, whereas our work is based on different time points at the sentence level. This could explain the different encoding performance in IFG.
> >
> > **Question4**
> >
> > RE24: We have now added standard error shading to all line plots.
> >
> > **Question5**
> >
> > RE25: RE25: We have now tuned down the statement of dLLMs in the introduction as: “Together, these results suggest that diffusion models may capture some aspects of the dynamics of human speech production in addition to autoregressive models.” We have also revised the first sentence of the conclusion as : “In conclusion, our study shows that dLLMs could potentially explain some aspects of human speech production.”
> > We have now included results in which the revelation order of words was fully randomized at each step. As expected, this procedure did not yield any meaningful alignment patterns (see Figure 12 in Appendix K). We also increased the number of denoising steps to 10 for longer production sentences (25-50 words), and the results continue to show that earlier step dLLM embeddings provide the best fit in the early time window of the left MTL/ITL. We did not examine a rightward-only revelation order, as no existing model operates in this manner. Consequently, comparing a non-functional “rightward-only” model to the brain would not provide an interpretable or meaningful baseline. We have also included data from comprehension sentences (5-25 and 25-50 word long), the results showed. Additionally, we have included results from fMRI data where participants passively listen to a 2-hour long audiobook (see Section 4.3 and Figure 14 in Appendix M). We believe that these additional analyses strengthened our conclusion on dLLMs’ better alignment during production.
> >
> > **Question6**
> >
> > RE26: We have now performed 4 ridge regressions with the 4 LLM embeddings independently at every layer and we averaged the correlation coefficients over all sensors and timepoints. We then reconducted the banded ridge regressions using the best layer embeddings from each LLM (see Figure 5 in Appendix D).
> >
> > **Question7**
> >
> > RE27: The labels of the y-axes for Figure 3b and 3c were actually the same as the figure title. We have now added them to the y-axes.

---

> > > ### Author Response · Authors · 2025-11-25
> > >
> > > **Question8**
> > >
> > > RE28: The percentage measures were computed as follows: for each generation step, we first calculated the proportion of each POS tag relative to the total number of tags at that step across all sentences, and then averaged these proportions across the two dLLMs and across the aLLMs, respectively. For example, if in Step 1 the dLLM ordering yields 50 nouns and the aLLM (i.e., sequential) ordering yields 40 nouns, and there are 100 total words in Step 1 across all sentences, then the percentage of nouns would be 50% for the dLLM and 40% for the aLLM.
> > >
> > > **Question9**
> > >
> > > RE29: We have now added a table with model specifics in Appendix C.
> > >
> > > **Question10**
> > >
> > > RE30: We originally chose five steps because our initial sentence-length range was 5–25 words; using five steps ensured that each step contained a reasonable number of words (neither too few nor too many). After adding results from longer sentences (25–50 words), we increased the number of denoising steps to ten. Even with this extended range and finer temporal resolution, we still observed stronger model fit for the first-step dLLM embeddings in the early time window of the MTL/ITL (see Figure 13 in Appendix L). This suggests that the effect observed with 5–25 word sentences and five denoising steps generalizes to longer sentences when more steps are used.
> > >
> > > **Question11**
> > >
> > > RE31: Goldstein et al. (2025) showed that language embeddings reduced to two dimensions using t-SNE exhibit clear POS clusters. In contrast, we did not examine whether the LLM embeddings themselves contain POS structure. Instead, we quantified how many tokens of each POS tag appear at each generation step—a measure based solely on percentage of tag counts rather than on embedding representations. The 3D embeddings shown in Figure 3a illustrate only the distribution of embeddings across the five steps for aLLMs and dLLMs; they do not contain any information about the POS categories those embeddings represent.

---

### Official Review · Reviewer_rYyJ · 2025-10-29

**Soundness:** 2
**Presentation:** 2
**Contribution:** 2
**Rating:** 2
**Confidence:** 4

**Summary:**

This paper presents an investigation into the cognitive plausibility of diffusion-based large language models (dLLMs) as a model for human speech production. The authors propose that the iterative, global refinement process of dLLMs may better capture the pre-articulatory planning phase of speech. To test this hypothesis, the authors correlate the internal representations from dLLMs (LLaDA-8B, Dream-7B) and their aLLM counterparts (LLaMA3-8B, Qwen2.5-7B) with electrocorticography (ECoG) data from four patients engaged in naturalistic conversation. They extract embeddings from five progressive "steps" for both model classes. For dLLMs, these steps are defined by a novel "greedy confidence-based token revelation" algorithm, which iteratively unmasks the tokens of a known target sentence in order of the model's confidence. The authors conclude that these results support iterative refinement as a plausible neural mechanism for human speech planning

**Strengths:**

1. The idea of investigating the connection between diffusion language models and human brain response is novel and interesting.

**Weaknesses:**

1. The biggest problem of this paper is that the authors fail to provide any neurological basis & insight of why dLLMs can align to human brain responses and the connection & differences between aLLMs and dLLMs in aligning to brain response.
The research method is common and has been widely used in previous work. So only the analysis of dLLMs-brain alignment becomes an interesting point.
However, this authors just provide brain encoding performance and do not conduct any further analysis.
Although the authors mentioned "These results suggest that the brain’s production system may function more like an iterative refinement process than a strictly left-to-right generator." Such claim is arbitrary and lacks supporting material.
This weakness greatly reduces the contribution of this paper.

2. The authors fall short in investigating the alignment between dLLMs and brain response in the context of other cognitive signals (e.g. fmri). Besides, the study's experiment on ECoG data only involves four patients. Given the known variability in electrode placement and individual pathology, N=4 is an insufficient sample size to make broad claims. Previous studies used fmri dataset with hundreds of subjects.

**Questions:**

Please refer to the weaknesses.

---

> ### Author Response · Authors · 2025-11-25
>
> We greatly appreciate your time and effort in reviewing our manuscript, and we look forward to engaging with you in a constructive and thoughtful discussion aimed at further improving the quality of the work. Below is our point-by-point responses to the weaknesses and questions:
>
> **Weakness1**
>
> RE14: As you noted, the model–brain alignment approach has been widely adopted in prior work. The underlying rationale is that, by comparing how well different models align with neural data, we may infer which computational mechanisms instantiated by these models best approximate the brain’s representations or processes, which still remain largely unknown in cognitive neuroscience of language due to lack of animal model. This logic has been used extensively in previous studies (e.g., Caucheteux et al., 2023; Gao et al., 2025; Goldstein et al., 2022; 2025; Schrimpf et al., 2021; Toneva & Wehbe, 2019; Antonello et al., 2024; Jain & Huth, 2018). To my knowledge, none of these studies provides causal evidence that the brain operates in an autoregressive manner; rather, they rely on correlational model–brain comparisons. We would therefore greatly appreciate any concrete suggestions regarding what types of evidence, beyond model comparison, prior studies have offered that support mechanistic claims about brain function.
>
> **Weakness2**
>
> RE15: Although our dataset includes only four patients, each participant was recorded continuously over multiple days (24/7), yielding approximately 100 hours of neural data in total. This level of within-subject sampling is not comparable to typical fMRI datasets, which usually contain only 1–2 hours of data per subject. We therefore believe that the dataset provides sufficient statistical power. Indeed, many prior studies have included only 3–4 participants but with extensive neural data per individual (e.g., Goldstein et al., 2022, 2024, 2025; Huth et al., 2016; Tang et al., 2023). Another reason we did not use large open-access fMRI datasets with hundreds of participants is that diffusion-based mechanisms are most relevant to language production, given that diffusion models are generative. In contrast, language comprehension proceeds incrementally as words are heard, and is therefore more naturally aligned with autoregressive processing. That said, as Reviewer UqtJ also suggested, we have now included results from an open fMRI dataset of naturalistic listening consisting of approximately two hours of story comprehension data from 49 English participants (Li et al., 2022). This additional analysis is now reported in Section 4.3 and Appendix M of the revised manuscript.
>
> * Goldstein, A., Grinstein-Dabush, A., Schain, M., Wang, H., Hong, Z., Aubrey, B., Nastase, S. A., Zada, Z., Ham, E., Feder, A., Gazula, H., Buchnik, E., Doyle, W., Devore, S., Dugan, P., Reichart, R., Friedman, D., Brenner, M., Hassidim, A., … Hasson, U. (2024). Alignment of brain embeddings and artificial contextual embeddings in natural language points to common geometric patterns. Nature Communications, 15(1), 2768.
> * Goldstein, A., Wang, H., Niekerken, L., Schain, M., Zada, Z., Aubrey, B., Sheffer, T., Nastase, S. A., Gazula, H., Singh, A., Rao, A., Choe, G., Kim, C., Doyle, W., Friedman, D., Devore, S., Dugan, P., Hassidim, A., Brenner, M., … Hasson, U. (2025). A unified acoustic-to-speech-to-language embedding space captures the neural basis of natural language processing in everyday conversations. Nature Human Behaviour, 9(5), 1041–1055.
> * Goldstein, A., Zada, Z., Buchnik, E., Schain, M., Price, A., Aubrey, B., Nastase, S. A., Feder, A., Emanuel, D., Cohen, A., Jansen, A., Gazula, H., Choe, G., Rao, A., Kim, C., Casto, C., Fanda, L., Doyle, W., Friedman, D., … Hasson, U. (2022). Shared computational principles for language processing in humans and deep language models. Nature Neuroscience, 25(3), Article 3.
> * Li, J., Bhattasali, S., Zhang, S., Franzluebbers, B., Luh, W.-M., Spreng, R. N., Brennan, J. R., Yang, Y., Pallier, C., & Hale, J. (2022). Le Petit Prince multilingual naturalistic fMRI corpus. Scientific Data, 9(1), Article 1.
> * Huth, A. G., de Heer, W. A., Griffiths, T. L., Theunissen, F. E., & Gallant, J. L. (2016). Natural speech reveals the semantic maps that tile human cerebral cortex. Nature, 532(7600), 453–458.
> * Tang, J., LeBel, A., Jain, S., & Huth, A. G. (2023). Semantic reconstruction of continuous language from non-invasive brain recordings. Nature Neuroscience, 26, 1–9.

---

### Official Review · Reviewer_sif2 · 2025-10-30

**Soundness:** 2
**Presentation:** 3
**Contribution:** 3
**Rating:** 4
**Confidence:** 4

**Summary:**

This work compares large language models based on diffusion vs. autoregressive processes to predict human brain activity during speech production, as recorded by electrocorticography. The authors show that, with their settings, diffusion LLMs yield better brain scores than autoregressive LLMs, and conclude that the iterative refinement strategy is thus a better neural model of speech planning in humans.

**Strengths:**

- To my knowledge, this is the first study that investigates the brain alignment of diffusion large language models.
- While many papers study natural language processing during listening or reading, there are not many works on the production side, which is an interesting complementary research question.
- The authors compare aLLM and dLLM with similar versions of the same architecture, making the comparison on comparable grounds.

**Weaknesses:**

The work is interesting and novel but not entirely convincing due to some methodological issues.

- An important potential issue is related to how the sentence embeddings are created.
This is a technical point that is important to assess the comparison between the aLLM and the dLLM. Here, as far as I understood, the sentence embeddings are created by averaging over all tokens of the sentence (the one corresponding to the production data). In non-autoregressive LLMs, each token at the output has the knowledge of the whole sentence, whereas for an autoregressive LLM, for a given token, only past words are seen. Averaging over all tokens in a non-autoregressive setting feels intuitive, but for an autoregressive LLM, averaging all the tokens, although understandable, is not necessarily the obvious choice. As the sentence gets built, cumulatively averaging the tokens might lead to blurring the embedding. Instead, one might want to use the last token in the sentence. This is an empirical question in the end, but it needs to be tested, as it might result in an unfair comparison between aLLM and dLLM, favoring the non-autoregressive representations. Relatedly, this might explain the very low brain scores for aLLM (Fig. 4a and 6; see notably IFG), which are very weak, and seem inconsistent with the existing literature. Likewise, this crucial point about how the sentence embedding is created could explain the negative results on Fig. 3a, with more blurring leading to less separated dimensions.

- Although partially acknowledged as a limitation in the conclusion, the a priori choice of a layer (layer 20 here) could be hindering some differences between aLLM and dLLM, as the best layer is not necessarily the same for both variants. Although the rationale behind the choice of the best layer makes sense, this choice is questionable. First, previous work has shown that there is large variations in the relative depth of the best layer between model families (see e.g., Antonello et al., 2023, Fig 1c), and also as a function of the performance/model size: the larger the model, the earlier the layer in terms of its relative depth (Hong et al., 2024).
It would be interesting to compare layer by layer the brain alignment of both variants (autoregressive and diffusion), which would provide a better ground for a fair comparison between the two aLLMs and the dLLMs. This seems straightforward to investigate, without much more work (albeit more compute).

- Throughout the paper, there are several claims that the results concern specifically the production process (eg "iterative refinement as a plausible neural mechanism of human speech planning", "these results highlight diffusion models as a cognitively plausible framework for capturing the dynamics of human speech production", "dLLMs capture neural dynamics of human speech production in ways that qualitatively differ from aLLMs"). It is not obvious from the results that this is the case. It would be interesting to apply the same analysis to the "comprehension" part of the prompt, which would serve as a baseline to compare the production results with.

- Concerning the analyses presented in Fig. 3c and 3d, as well as lines 336 and following, I am not sure I have fully understood. The aLLM results do not seem to depend on any model, simply on the sequential left-to-right nature of the words, contrary to other results, such as the one involving embeddings. If this is true, then the reporting is misleading: it is just a comparison with a baseline, that compares with natural left-to-right ordering in English, not a property of the language models themselves.

Minor comments/typos.

- The Introduction and Related Work might leave an unfamiliar reader with the feeling that only autoregressive LLMs have been used to predict brain activity, whereas many works have used non-autoregressive LLMs (typically BERT) to model natural language processing in the human brain (see e.g., Toneva & Wehbe, 2019; Sat et al., 2019; Schrimpf et al., 2021; Caucheteux et al., 2021; Lamarre et al., 2022). Those references focus on language comprehension, but even for production, Goldstein et al. (2025) use the representation obtained by the encoder of the Whisper model, which is non-autoregressive.
Although this is probably known by the authors, it might be worth discussing these works a bit better.

- typos: Fig. 2 average vs. averaged

**Questions:**

- How exactly are the sentence embeddings created? And how the results depend on this choice? In particular, using the last token for the sentence representation might be a better choice for the autoregressive LLMs (see Weaknesses section).

- How was the threshold for sentence length chosen? A look at the distribution of sentence length, Fig. 5a, does not provide an obvious answer such as a natural boundary. It seems that the specific range that is chosen by the authors might favor the dLLM compared to the aLLM. Imagine a very long sentence; it feels like at some point the left-to-right sequential order should be important for the alignment with brain activity. It would be fairer to use a larger range of the distribution, and then study how the two approaches compare as a function of sentence length. This should result in a cleaner and more insightful way to compare the two approaches, and might provide insights on how the two processes might complement each other on different time scales.

- What are the positions of the tokens at each step during token revelation in dLLMs?
As a supplementary figure, it would be useful to see the successive positions of the tokens that are chosen. The case of a left-to-right reveal would amount to the strategy provided by the aLLM. Such an analysis would be useful to see if there are specific biases in the working of the dLLM, and the departure from the autoregressive processes.

---

> ### Author Response · Authors · 2025-11-25
>
> Thank you very much for your constructive suggestions, which have greatly helped improve our work. Below are our point-by-point responses to the identified weaknesses and questions:
>
> **Weakness1**
>
> RE6: Our decision to average the aLLM embeddings across the five generation steps was made actually to ensure a fair comparison with the dLLM embeddings, which are themselves averaged across all tokens at each denoising step. In dLLMs, unrevealed tokens contribute “noisy” representations that gradually sharpen over steps; averaging captures this evolving whole-sentence representation. Using last-token embeddings for aLLMs but averaged embeddings for dLLMs would therefore introduce an asymmetry in the comparison. We also computed the correlation between last-token embeddings and averaged-token embeddings for each sentence at each step in the aLLMs. Although the correlation decreases as more tokens are included in the average at later steps, all correlations remain in the range of 0.6–0.8 (see Figure 6 in Appendix F), suggesting that the averaged embeddings still reasonably approximate the last-token embeddings. Given this empirical result and the need for methodological symmetry with dLLMs, we retained the averaged-embedding approach.
>
> We agree that the brain scores appear lower than those reported in prior studies (e.g., Goldstein et al., 2025). This is because our main analyses used banded ridge regression with all four LLMs included simultaneously, and we reported only the unique variance (r values) independently explained by each model. Given that the two aLLM embeddings and the two dLLM embeddings are highly correlated, the unique variance attributed to each individual model is necessarily reduced. Additionally, our encoding analyses were performed at the sentence level rather than the word level as in Goldstein et al. (2025), resulting in substantially fewer trials for training the encoding models. This reduced trial count may have contributed to the lower brain scores observed in our study.
>
> **Weakness2**
>
> RE7: We agree with your points and have now computed the encoding performance of both aLLMs and dLLMs across all layers. Due to limited time and computing resources, we performed this analysis only for the full sentence embedding (step 5) using averaged token embeddings. Figure 5 in Appendix D shows the averaged encoding performance across all sensors and timepoints. For comprehension sentences, the best-performing layers for LLaMA, LLaDA, Qwen, and Dream were Layer 15, 12, 13, and 18, respectively. For production sentences, the corresponding best layers were 30, 27, 14, and 27. Overall, the best layers for comprehension tend to occur earlier than those for production across all four models. We have updated all encoding models to use the empirically identified best layers.
>
> **Weakness3**
>
> RE8: We agree with your point and have now conducted the same analyses using comprehension sentences. The results showed that, for comprehension, later-step embeddings from aLLMs consistently yield higher encoding performance across all ROIs (see Figure 4c). dLLM embeddings do not exhibit a clear alignment pattern during comprehension. This suggests that the diffusion–brain alignment observed in our main analyses is specific to speech production rather than a general property of language processing.
>
> **Weakness4**
>
> RE9: Yes, the aLLM progression steps correspond to the natural left-to-right generation process divided into five steps. This follows directly from the definition of autoregressive models, in which continuations unfold word by word. In this case, aLLMs can be considered as the “baseline”. Importantly, we used aLLMs to extract embeddings as the sentence incrementally unfolds from left to right, in contrast to dLLMs, where the sentence is revealed in a non-linear manner across denoising steps.
>
> **Minor comments/typos**
>
> RE10: We mostly focused on literature using recent LLMs and GPT models while we admit that there are many work using BERT.  We have updated the introduction to included these references in “Related Work” on p. of the reivsed manuscript: Although some non-autoregressive models (e.g., BERT, Whisper) have also been used to investigate human language comprehension (Toneva & Wehbe, 2019; Schrimpf et al.,2021; Caucheteux et al., 2021; Lamarre et al., 2022; Schwartz et al., 2019) and production (Gold-stein et al., 2025), today’s state-of-the-art LLMs are largely autoregressive. Correspondingly, the mainstream view in psycholinguistics holds that readers and listeners actively anticipate upcoming words in an autoregressive manner (DeLong et al., 2005; Kutas Hillyard, 1980).
>
> We have fixed the typo. Thank you very much!

---

> ### Author Response · Authors · 2025-11-25
>
> **Question1**
>
> RE11: For aLLMs, we divided the sentence into five progressively longer partial inputs, containing roughly the first 20%, 40%, 60%, 80%, and 100% of words. For example, for the sentence “So maybe you did have something which is infinitely better than zero,” the five partial inputs would be: “So maybe”, “So maybe you did”, “So maybe you did have something”, “So maybe you did have something which is infinitely”, “So maybe you did have something which is infinitely better than zero”. Within each step, we averaged the token embeddings across all words included in that segment to obtain a single stage-level representation (see Figure 2a, upper panel). The resulting layer-wise representations consist of five progressive embeddings per sentence per model layer, capturing how meaning representations evolve as linguistic context accumulates. For dLLMs, we first obtained the diffusion order of the original sentence (e.g., “have is So which something maybe did you zero infinitely better than”), and then extracted the embeddings for each of the five denoising steps by averaging the token embeddings across the entire sentence at that step (see Figure 2a, lower panel). Importantly, the hidden states of unrevealed tokens continue to update at every step and gradually converge toward their final representations, allowing the denoising trajectory to capture the progressive refinement characteristic of diffusion-based generation. In this way, the stepwise embeddings for both aLLMs and dLLMs reflect how meaning is represented as it unfolds—either through left-to-right generation or through iterative denoising.
>
> **Question2**
>
> RE12: We selected the 5–25 word range because it reflects typical utterance lengths in everyday communication. Sentences approaching 25 words (e.g., “hard for me to swallow nine grams of protein the protein when you're not eating protein turns into carbohydrates plenty of it's plenty of cranberries”) already tend to include many repetitions, hesitations (e.g., “um”), or repairs, and extending far beyond this range (e.g., 25–50 words) often results in unusually long and less fluent utterances. However, we agree that sentence length may influence aLLM–dLLM comparisons. Therefore, we have now added analyses using comprehension and production sentences between 25–50 words. The results show that earlier-step dLLM embeddings continue to better predict neural responses in MTL/ITL and motor cortex during production (see Figure 13 in Appendix L), consistent with the findings from the 5–25 word range .
>
> **Question3**
>
> RE13: Table 1 in the original manuscript (now moved to Table 3 in Appendix H) illustrated five example sentences showing the words revealed at each step for the dLLMs. We have now additionally quantified the positional differences between the positions of words from dLLMs and their original positions in the sentence. The results show that words appearing early in the sentence (original Step 1) are unlikely to be placed in the final step (Step 5), and words originally occurring at the end of the sentence are more likely to appear in the later steps. However, words from the first step are distributed relatively evenly across Steps 2–4, indicating flexibility in the intermediate denoising stages (see Figure 7 in Appendix H).

---

> > ### Comment · Reviewer_sif2 · 2025-11-27
> >
> > Thank you for your work on the rebuttal. It has answered many important issues, and I also appreciate your responses to the other reviewers. The new analyses clarify certain points and also bring some new interesting results.
> > Overall, I believe this has indeed improved the work and I will be happy to increase my score.
> >
> > **Weakness 1**
> >
> > Although I understand the rationale put forward regarding what is fair, I still believe it is actually not obvious which method is fairer, and providing both would help the reader decide. You could provide the equivalent of Fig. 4
> > as a supplement to Fig. 6 in Appendix F, as the same correlation between embeddings could lead to opposite results when turning to brain scores. Could you provide such a Figure? I would be curious also to see the equivalent of Fig. 3a.
> >
> > **Weakness 3**
> >
> > I really appreciate the new results on comprehension vs. production (Sec. 4.2.2, Fig. 4, 13, and Appendix I). In Fig. 4 (and the new Fig. 13), it would be easier to compare between conditions (autoregressive vs. diffusion, comprehension vs. production) if the graphs shared the same limits for the y-axes.
> >
> > **Question 2**
> >
> > Previous Appendix B showing the summary statistics of production sentences has disappeared in the revised version. I think it is informative and should stay there: why have you removed it?

---

> > > ### Author Response · Authors · 2025-11-28
> > >
> > > Thank you so much for recognizing the work we put into the revisions. We would be grateful if you could update your score to reflect the improvements.
> > >
> > > **Weakness1**
> > >
> > > We have now added the PCA analyses and encoding results derived from last-token aLLM embeddings (equivalent to Figures 3a and 4) to Appendix N. Please note that we could not insert additional figures into the earlier appendices at this stage, as doing so would disrupt the figure numbering referenced in our responses to other reviewers.
> > >
> > > **Weakness3**
> > >
> > > We originally used different y axes because we want to illustrate how step-wise encodings differed within each ROI rather than across ROIs. Following your suggestion, we have now standardized all plots to use the same y-axis range (Figure 4 and Figure 13). As expected, the curves now lie quite close to one another, which makes the step-wise patterns more difficult to distinguish.
> > >
> > > **Question2**
> > >
> > > We removed this plot earlier because our analyses focused solely on 5–25 word production sentences with a 50-word cutoff, and the goal was simply to show what proportion of all production sentences fell within that range. Now that we have expanded the analyses to include 5–50 word sentences for both comprehension and production, we felt that visualizing the distribution of all sentences would be less informative. Nevertheless, following your suggestion, we have reinstated the plot as Figure 17 in Appendix O.

---

### Official Review · Reviewer_V6ig · 2025-10-30

**Soundness:** 3
**Presentation:** 4
**Contribution:** 3
**Rating:** 6
**Confidence:** 4

**Summary:**

The paper aims to test a hypothesis on whether the human brain conducts iterative refinement of language production during pre-articulatory planning.

**Strengths:**

The paper is the first to test, via ecog analysis using diffusion modeling, whether the human brain may conduct iterative refinement of language production during pre-articulatory planning.  I found the paper’s goal to be interesting, and supported by  experimentation.

**Weaknesses:**

- I have a general hesitation toward papers that take one particular modeling approach and then compare its alignment to another model with the brain data. The authors should be even more careful in stating the implications of these findings. If a model fits better with neural data, it does not yet mean that this model has anything to do with the brain representation or processes — it simply means that the model has a better fit in the studied aspect than some other model, but there is no mechanistic evidence of the brain functionality.

- The outputs of the models with are different than standard autoregressive models, but that could also be simply because autoregressive are *autoregressive* and output the most likely outcomes given the t-0..k observations. Diffusion has a different approach based on denoising, and it is natural that the inference steps treat the likelihoods differently. Why would that be strongly linked to the brain function? Couldn't this simply be because the model works differently?

- Some key recent papers on (semantic) language reconstruction (see e.g. Nature and Nature Communications / Communications Biology, but also CS venues) are missing and should be discussed.

**Questions:**

- I would have liked to see control conditions with permutated data that breaks the ties between the neural recordings and the language, i.e., retains the distribution of data, but has random connections to the language tokens. Sometimes a model follows a kind of mode collapse average outputs, and this explains part of the results. Also, significance testing should be done for this according to the keep-it-maximal strategy.

- I would have liked to see an analysis on generative performance with more controlled scenarios, like predictions on "unlikely" outcomes for which either autoregressive nor other model relying strongly on sequential information would not give high likelihoods. It may well be that this would have led to different conclusions. In summary, the present evidence is sufficient, but not comprehensive, for the claims on brain functionality.

**Details Of Ethics Concerns:**

Brain data, but with public dataset

---

> ### Author Response · Authors · 2025-11-25
>
> Thank you very much for your your insightful feedback and constructive suggestions. We have carefully addressed all the points
> you raised, and we hope that you will find this version much strengthened. Below is our point-by-point responses to the weaknesses and questions:
>
> **Weakness1**
>
> RE1: We agree with your concern and have added a discussion to clarify the scope and interpretation of our findings: “Higher model–brain alignment does not necessarily imply that a model definitely implements the same mechanisms as the human brain. Rather, a better model fit only indicates that the model captures certain statistical or representational properties that are more predictive of the neural responses within the specific task and dataset studied”.
>
> **Weakness2**
>
> RE2: Our rationale for comparing diffusion LLMs with brain data is that prior model–brain alignment studies have focused mostly on autoregressive LLMs and have suggested that human language processing is also autoregressive. However, recent diffusion LLMs, despite relying on a fundamentally different prediction objective, are likewise capable of generating plausible sentence continuations. If so, this would suggest that human language production may not rely solely on autoregressive mechanisms. Importantly, this is an empirical question that can be directly tested. We have added the motivation in the Introduction to make the rationale more explicit.
>
> **Weakness3**
>
> RE3: The following papers from Nature Neuroscience/Nature Human Behavior/Nature Communications/Communications Biology/NeurIPS have already been included in the manuscript:
> * Charlotte Caucheteux, Alexandre Gramfort, and Jean-R´emi King. Evidence of a predictive coding hierarchy in the human brain listening to speech. Nature Human Behaviour, 7(3):430–441, 2023.
> * Charlotte Caucheteux and Jean-R´emi King. Brains and algorithms partially converge in natural language processing. Communications Biology, 5(1), 2022.
> * Ariel Goldstein, Zaid Zada, Eliav Buchnik, Mariano Schain, Amy Price, Bobbi Aubrey, Samuel Nastase, Amir Feder, Dotan Emanuel, Alon Cohen, et al. Shared computational principles for language processing in humans and deep language models. Nature Neuroscience, 25(3):369–380, 2022.
> * Ariel Goldstein, Haocheng Wang, Leonard Niekerken, Mariano Schain, Zaid Zada, Bobbi Aubrey, Tom Sheffer, Samuel Nastase, Harshvardhan Gazula, Aditi Singh, Aditi Rao, Gina Choe, Catherine Kim, Werner Doyle, Daniel Friedman, Sasha Devore, Patricia Dugan, Avinatan Hassidim, Michael Brenner, Yossi Matias, Orrin Devinsky, Adeen Flinker, and Uri Hasson. A unified acoustic-to-speech-to-language embedding space captures the neural basis of natural language processing in everyday conversations. Nature Human Behavior, 9(5):1041–1055, 2025.
> * Shailee Jain and Alexander Huth. Incorporating context into language encoding models for fMRI. In Advances in Neural Information Processing Systems, 2018.
> * Richard Antonello, Aditya Vaidya, and Alexander G. Huth. Scaling laws for language encoding models in fmri. In Advances in Neural Information Processing Systems, 2024.
>
> We have also added the following papers from Nature Communications, EMNLP and ICML in the revised version:
> * Ariel Goldstein, Avigail Grinstein-Dabush, Mariano Schain, Haocheng Wang, Zhuoqiao Hong, Bobbi Aubrey, Samuel Nastase, Zaid Zada, Eric Ham, Amir Feder, Harshvardhan Gazula, Eliav Buchnik, Werner Doyle, Sasha Devore, Patricia Dugan, Roi Reichart, Daniel Friedman, Michael Brenner, Avinatan Hassidim, Orrin Devinsky, Adeen Flinker, and Uri Hasson. Alignment of brain embeddings and artificial contextual embeddings in natural language points to common geometric patterns. Nature Communications, 15(1):2768, 2024.
> * Refael Tikochinski, Ariel Goldstein, Yoav Meiri, Uri Hasson, and Roi Reichart. Incremental accumulation of linguistic context in artificial and biological neural networks. Nature Communications, 16(1):803, 2025.
> * Caucheteux, C., Gramfort, A., & King, J.-R. (n.d.). Model-based analysis of brain activity reveals the hierarchy of language in 305 subjects.
> * Charlotte Caucheteux, Alexandre Gramfort, Jean-Remi King Proceedings of the 38th International Conference on Machine Learning, PMLR 139:1336-1348, 2021
>
> We would greatly appreciate suggestions for additional key literature that we may have missed.

---

> > ### Author Response · Authors · 2025-11-25
> >
> > **Question1**
> >
> > RE4: Thank you very much for the suggestion. We have now added results from a control condition in which the ECoG data were permuted. In this control analysis, the regressors entered into the banded ridge regression were: the spectrogram, Whisper embeddings (last encoder layer, following prior literature; Goldstein et al., 2025), the best-performing layers of LLaDA, LLaMA, Dream, and Qwen at each denoising or autoregressive step for each sentence. The acoustic and speech features were included following Reviewer sif2’s suggestions, and we replaced the fixed layer 20 with the best-performing layer for each LLM in accordance with the suggestions from Reviewers sif2 and Gw42. The dependent ECoG data were permuted across sentences for all 15 timepoints, such that the embeddings no longer corresponded to the true sentence-level neural responses. We did not observe any alignment for the acoustic, speech, aLLM, or dLLM predictors in this control condition (see Figure 10-11 in Appendix K).
> >
> > **Question2**
> >
> > RE5: Thank you for this insightful suggestion. We agree that performance should be evaluated on those involving highly “unlikely” continuations. Unfortunately, our current dataset consists exclusively of naturalistic conversations, which do not include experimentally manipulated or systematically constructed unlikely outcomes. That said, spontaneous conversational speech in our dataset does contain numerous hesitations, repairs, and speech errors (e.g., “really it feels a lot better, they took the, they took the thing out, the drain out.”), which are inherently unlikely and generally not produced by LLMs during free generation. Thus, we believe the naturalistic data already include many instances of low-probability or disfluent continuations that LLMs struggle to predict. We fully agree, however, that systematically probing generative performance under controlled improbable scenarios would strengthen the conclusions, and we have added this point as an important direction for future work in the revised Discussion.

---

> > ### Comment · Reviewer_V6ig · 2025-11-26
> > **Comment**
> >
> > Thank you for your responses.
> >
> > Some work that I hope you are aware of (also for the experimental designs and controls):
> >
> > https://www.nature.com/articles/s41593-023-01304-9
> > https://arxiv.org/abs/2412.17829
> > https://www.nature.com/articles/s42003-025-07731-7

---

> > > ### Author Response · Authors · 2025-11-28
> > >
> > > Thank you for providing these references. We are very familiar with this line of work; however, our study focuses on encoding rather than decoding, whereas these papers primarily use decoding approaches. That is why we had not included them previously. We have now added these references to the revised Introduction.

---

### Meta-Review · Area_Chair_Gj7x · 2026-01-13

**Summary:**

This paper correlates language models' intermediate representations with ECoG signals of humans during speech. Experiments found that diffusion LMs better explains neural variance, which supports "diffusion LMs as a plausible neural mechanism of human speech planning". However, after the rebuttal, there are several concerns that remain: absolute performance remains low compared to prior ECoG encoding work; evaluation focuses on mid-size models and might not generalize to different architectures and modalities; and most importantly, this work is more of a correlational interpretation instead of mechanistic interpretation, lacking support for the core cognitive claim. Therefore, I'm not recommending its acceptance.

**Reviewer Concerns:**

1. The concerns about lacking baselines using established comprehension/production mapping studies are mitigated by new baselines.
2. The concerns about lack of controls are addressed by adding a new fMRI control analysis.
3. The concerns about variance partitioning is addressed.

For concerns not fully addressed, see above summary.

**Reviewer Scores:**

1. V6ig: unlikely to change given that their score is already positive and that the reviewer has replied to author rebuttal.
2.sif2: the concern about lower absolute performance compared to prior works remains, and the concern about averaging embeddings instead of alternatives is not directly addressed by experiments (although indirectly through similarity), so likely keeps their score.
3. rYyJ: although the review is short, the concerns seem valid. The concern about fMRI control is mitigated, but the concern about the lack of insights and the work just being a correlational analysis is still outstanding. Therefore, it is likely that this reviewer might slightly increase their score from 2 but still leaning negative.
4. Gw42: limited experiments and small models not really addressed. Likely keeps their score.
5. UqtJ: explicitly stated that they will keep their score while decreasing confidence.

---

### Decision · Program_Chairs · 2026-01-26

Reject